# New insights into anatomical connectivity along the anterior–posterior axis of the human hippocampus using *in vivo* quantitative fibre tracking

Marshall A Dalton[1,2,3]*, Arkiev D'Souza[2,4,5], Jinglei Lv[1,2], Fernando Calamante[1,2,5]

[1]School of Biomedical Engineering, Faculty of Engineering, The University of Sydney, Sydney, Australia; [2]Brain and Mind Centre, The University of Sydney, Sydney, Australia; [3]School of Psychology, Faculty of Science, The University of Sydney, Sydney, Australia; [4]Faculty of Medicine and Health Translational Research Collective, The University of Sydney, Sydney, Australia; [5]Sydney Imaging, University of Sydney, Sydney, Australia

**Abstract** The hippocampus supports multiple cognitive functions including episodic memory. Recent work has highlighted functional differences along the anterior–posterior axis of the human hippocampus, but the neuroanatomical underpinnings of these differences remain unclear. We leveraged track-density imaging to systematically examine anatomical connectivity between the cortical mantle and the anterior–posterior axis of the *in vivo* human hippocampus. We first identified the most highly connected cortical areas and detailed the degree to which they preferentially connect along the anterior–posterior axis of the hippocampus. Then, using a tractography pipeline specifically tailored to measure the location and density of streamline endpoints within the hippocampus, we characterised where these cortical areas preferentially connect within the hippocampus. Our results provide new and detailed insights into how specific regions along the anterior–posterior axis of the hippocampus are associated with different cortical inputs/outputs and provide evidence that both gradients and circumscribed areas of dense extrinsic anatomical connectivity exist within the human hippocampus. These findings inform conceptual debates in the field and emphasise the importance of considering the hippocampus as a heterogeneous structure. Overall, our results represent a major advance in our ability to map the anatomical connectivity of the human hippocampus *in vivo* and inform our understanding of the neural architecture of hippocampal-dependent memory systems in the human brain.

*For correspondence: marshall.dalton@sydney.edu.au

## Editor's evaluation

This is an historical paper that is methodologically exceptional that offers new insights into the heterogeneity of human hippocampal anatomical pathways. Coming at a time when functional studies and theoretical papers are recognizing that this heterogeneity is of critical importance in furthering our understanding of hippocampal function, this paper will provide a nice guide for researchers in their ongoing hypothesis testing. Congratulations on an invaluable contribution!

## Introduction

There is long-standing agreement that the hippocampus is essential for supporting episodic long-term memory (*Scoville and Milner, 1957*) and facilitating spatial navigation (*Maguire et al., 2006*;

**eLife digest** The brain allows us to perceive and interact with our environment and to create and recall memories about our day-to-day lives. A sea-horse shaped structure in the brain, called the hippocampus, is critical for translating our perceptions into memories, and it does so in coordination with other brain regions. For example, different regions of the cerebral cortex (the outer layer of the brain) support different aspects of cognition, and pathways of information flow between the cerebral cortex and hippocampus underpin the healthy functioning of memory.

Decades of research conducted into the brains of non-human primates show that specific regions of the cerebral cortex anatomically connect with different parts of the hippocampus to support this information flow. These insights form the foundation for existing theoretical models of how networks of neurons in the hippocampus and the cerebral cortex are connected. However, the human cerebral cortex has greatly expanded during our evolution, meaning that patterns of connectivity in the human brain may diverge from those in the brains of non-human primates.

Deciphering human brain circuits in greater detail is crucial if we are to gain a better understanding of the structure and operation of the healthy human brain. However, obtaining comprehensive maps of anatomical connections between the hippocampus and cerebral cortex has been hampered by technical limitations. For example, magnetic resonance imaging (MRI), an approach that can be used to study the living human brain, suffers from insufficient image resolution.

To overcome these issues, Dalton et al. used an imaging technique called diffusion weighted imaging which is used to study white matter pathways in the brain. They developed a tailored approach to create high-resolution maps showing how the hippocampus anatomically connects with the cerebral cortex in the healthy human brain. Dalton et al. produced detailed maps illustrating which areas of the cerebral cortex have high anatomical connectivity with the hippocampus and how different parts of the hippocampus preferentially connect to different neural circuits in the cortex. For example, the experiments demonstrate that highly connected areas in a cortical region called the temporal cortex connect to very specific, circumscribed regions within the hippocampus.

These findings suggest that the hippocampus may consist of different neural circuits, each preferentially linked to defined areas of the cortex which are, in turn, associated with specific aspects of cognition. These observations further our knowledge of hippocampal-dependant memory circuits in the human brain and provide a foundation for the study of memory decline in aging and neurodegenerative diseases.

O'Keefe and Nadel, 1978). The hippocampus has more recently been linked with other roles including imagination of fictitious and future experiences (*Hassabis et al., 2007*; *Addis et al., 2007*), visuospatial mental imagery (*Dalton et al., 2018*), visual perception (*McCormick et al., 2021*; *Lee et al., 2012*), and decision-making (*McCormick et al., 2016*). It is a complex structure containing multiple subregions (referred to as subfields) including the dentate gyrus (DG), cornu ammonis (CA) 4-1, subiculum, presubiculum, and parasubiculum. Accumulating evidence suggests that different hippocampal subfields are preferentially recruited during different cognitive functions (*Dalton et al., 2018*; *Dimsdale-Zucker et al., 2018*). Functional differences are also present along the anterior–posterior axis of the hippocampus (*Poppenk et al., 2013*; *Plachti et al., 2019*; *Brunec et al., 2018*; *Przeździk et al., 2019*; *Poppenk, 2020*; *Strange et al., 2014*) and its subfields (*Dalton et al., 2019b*; *Dalton et al., 2019a*). Despite recent advances in understanding functional differentiation within the hippocampus, much less is known about the neuroanatomical underpinnings of these functional differences in the human brain. A more detailed understanding of anatomical connectivity along the anterior–posterior axis of the human hippocampus is needed to better understand and interpret these functional differences.

Much of our knowledge regarding the anatomical connectivity of the human hippocampus is inferred from the results of tract-tracing studies in rodent and non-human primate brains. While this information has been fundamental to inform theoretical models of hippocampal-dependent memory function, recent investigations have highlighted potential differences between connectivity of the human and non-human primate hippocampus (*Zeineh et al., 2017*). This suggests that, in addition to evolutionarily conserved patterns of hippocampal connectivity, human and non-human primates may

also have unique patterns of connectivity. More detailed characterisations of human hippocampal connectivity are, therefore, essential to advance our understanding of the neural architecture that underpins hippocampal-dependent memory and cognition in the human brain.

Tract-tracing studies in rodents and non-human primates have revealed that the hippocampus is highly connected with multiple cortical areas. It is well established that the entorhinal cortex (EC) is the primary interface between the hippocampus and multiple brain regions (*Garcia and Buffalo, 2020*; *Witter et al., 2017*). Other medial temporal lobe (MTL) structures including the perirhinal (PeEc) and posterior parahippocampal (PHC) cortices also have direct anatomical connections with the hippocampus (*Aggleton and Christiansen, 2015*; *Yukie, 2000*; *Insausti and Muñoz, 2001*; *Agster and Burwell, 2013*) albeit to a lesser degree. However, direct cortico-hippocampal pathways are not confined to MTL cortices. Anterograde and retrograde labelling studies in non-human primates have revealed direct and reciprocal connections between the hippocampus and multiple cortical areas in temporal (*Yukie, 2000*; *Van Hoesen et al., 1979*), parietal (*Rockland and Van Hoesen, 1999*; *Ding et al., 2000*), and frontal (*Goldman-Rakic et al., 1984*; *Barbas and Blatt, 1995*) lobes. These detailed investigations show that specific cortical areas preferentially connect with circumscribed portions along the anterior–posterior axis of hippocampal subfields. For example, retrograde labelling studies in the macaque reveal that the retrosplenial cortex (RSC) in the parietal lobe has direct connectivity with posterior portions of the presubiculum while area TE in the inferior temporal lobe displays preferential connectivity with posterior portions of the CA1/subiculum transition area (*Insausti and Muñoz, 2001*). While patterns of cortico-hippocampal connectivity such as these have been observed in the non-human primate brain, we know less about these patterns in the human brain. It is important to note that nomenclature relating to the long axis of the hippocampus differs between the human, non-human primate, and rodent literature. The human MRI literature predominantly refers to the anterior–posterior axis of the hippocampus. This corresponds to the rostral–caudal axis in the non-human primate literature and the ventral–dorsal axis in the rodent literature. For clarity, we exclusively use the term anterior–posterior throughout this article.

Detailed examination of structural connectivity (SC) of the human hippocampus has been difficult to pursue mainly due to the technical difficulties inherent to probing hippocampal connectivity *in vivo* using MRI. Limitations in both image resolution and fibre-tracking methods have precluded our ability to probe cortico-hippocampal pathways in a sufficient level of detail. Some researchers have partially circumvented these constraints by investigating blocks of *ex vivo* MTL tissue using high-field MRI scanners (*Augustinack et al., 2010*; *Coras et al., 2014*; *Beaujoin et al., 2018*) or novel methods such as polarised light microscopy (*Zeineh et al., 2017*). While these studies have provided important insights relating to MTL–hippocampal pathways, we have less knowledge regarding how the human hippocampus connects with more distant cortical areas. Detailed characterisations of anatomical connectivity between the hippocampal long-axis and broader cortical networks are needed to better understand functional heterogeneity within the hippocampus (*Olsen and Robin, 2020*).

Of the extant *in vivo* MRI studies that have examined SC of the human hippocampus, several have focused on the connectivity between the hippocampus and specific cortical or subcortical areas (*Dinkelacker et al., 2015*; *Wei et al., 2017*; *Arrigo et al., 2017*; *Rangaprakash et al., 2017*; *Edlow et al., 2016*) with a primary focus on disease states. To our knowledge, only one study has attempted to characterise the broader hippocampal 'connectome' in the healthy human brain. *Maller et al., 2019* used diffusion MRI (dMRI) data with multiple diffusion strengths and high angular resolution combined with track density imaging (TDI) to characterise SC between the whole hippocampus and cortical and subcortical brain regions. A quantitative analysis of streamline numbers (as a surrogate measure of connectivity) seeded from the whole hippocampus showed that the most highly connected brain regions were the temporal lobe followed by subcortical, occipital, frontal, and parietal regions. The primary focus of their study, however, was a description of six dominant white matter pathways that accounted for most cortical and subcortical streamlines connecting with the whole hippocampus. Together, these studies have provided an important glimpse into the complexity of human hippocampal SC but important gaps in our knowledge remain. Specifically, there has been no systematic examination of SC between the cortical mantle and the anterior–posterior axis of the human hippocampus, and we do not know where, within the hippocampus, specific cortical areas preferentially connect.

In typical fibre-tracking studies, we cannot reliably ascertain where streamlines would naturally terminate as they have been found to also display unrealistic terminations, such as in the middle of white matter or in cerebrospinal fluid (*Calamante, 2019*). While methods have been proposed to ensure more meaningful terminations (*Smith et al., 2012*), for example, with terminations forced at the grey matter–white matter interface (gmwmi), this approach is still not appropriate for characterising terminations within complex structures like the hippocampus. A key methodological advance of our approach was to remove portions of the gmwmi inferior to the hippocampus (where white matter fibres are known to enter/leave the hippocampus). This allowed streamlines to permeate the hippocampus in a biologically plausible manner. Importantly, we combined this with a tailored processing pipeline that allowed us to follow the course of streamlines within the hippocampus and identify their 'natural' termination points. These simple but effective methodological advances allowed us to map the spatial distribution of streamline 'endpoints' within the hippocampus. We further combined this approach with state-of-the-art tractography methods that incorporate anatomical information (*Smith et al., 2012*) and assign weights to each streamline (*Smith et al., 2015a*) to achieve quantitative connectivity results that more faithfully reflect the biological accuracy of the connection's strength (*Calamante, 2019*).

In this study, we aimed to systematically examine the patterns of SC between cortical brain areas and the anterior–posterior axis of the human hippocampus. We combined dMRI data from the Human Connectome Project (HCP), quantitative fibre-tracking methods, and a processing pipeline specifically tailored to study hippocampal connectivity with three primary aims: (i) to quantitatively characterise SC between the cortical mantle (focused on non-MTL areas) and the whole hippocampus; (ii) to quantitatively characterise how SC varies between cortical areas and the head, body, and tail of the hippocampus; and (iii) to use TDI combined with 'endpoint density mapping' to quantitatively assess, visualise, and map the spatial distribution of streamline endpoints within the hippocampus associated with each cortical area.

Our results represent a major advance in (i) our ability to map the anatomical connectivity of the human hippocampus *in vivo* and (ii) our understanding of the neural architecture that underpins hippocampal-dependent memory systems in the human brain. We provide fundamental insights into how specific cortical areas preferentially connect along the anterior–posterior axis of the hippocampus and identify where streamlines associated with a given cortical area preferentially connect within the hippocampus. These detailed anatomical insights will help fine-tune network connectivity models and will have an impact on current theoretical models of human hippocampal memory function. (A preliminary version of this work was presented at the 30th Annual Meeting of the International Society for Magnetic Resonance in Medicine; 17 May 2021).

## Results

We first characterised SC between the whole hippocampus and all cortical areas of the HCP Multi-Modal Parcellation (HCPMMP) (*Glasser et al., 2016*). The primary focus of this study was SC between the hippocampus and non-MTL cortical areas. We therefore focus on cortical areas outside of the MTL (information relating to MTL cortices is presented in Appendix 1, *Figure 1—figure supplement 1*, *Figure 2—figure supplement 1*, and *Supplementary files 1 and 2*).

### Which specific cortical areas most strongly connect with the whole hippocampus?

For brevity, we present results relating to the 20 cortical areas with the highest degree of SC with the whole hippocampus. Abbreviations for all cortical areas are defined in *Supplementary file 3*. The location of the most highly connected cortical areas are displayed in *Figure 2—figure supplements 2–4*. For the location of all other cortical areas, we refer the reader to the labelled Human Connectome Project Multi-modal Parcellation of Human Cerebral Cortex, which can be found at https://balsa.wustl.edu/sceneFile/Zvk4.

The hippocampus displayed the highest degree of SC with discrete cortical areas in temporopolar (areas TGv, TGd), inferolateral temporal (areas TF, TE2a, TE2p), medial parietal (areas RSC, ProS, POS1, POS2, DVT) dorsal and ventral stream visual (areas V3A, V6, FFC, VVC, VMV1, VMV2), and early visual (occipital) cortices (areas V1, V2, V3, V4). Results are summarised in *Figure 1A*, and *Table 1* lists

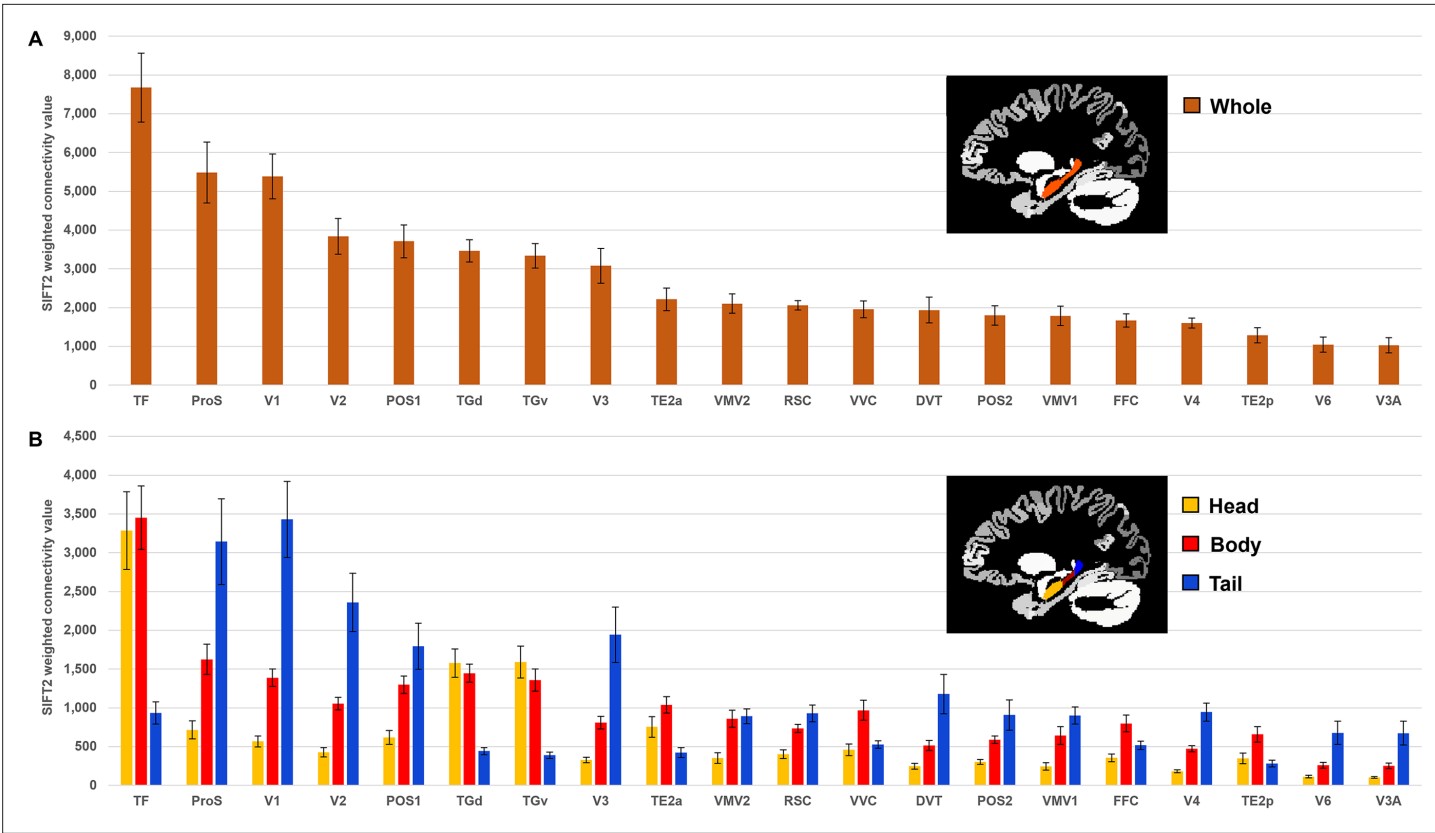

**Figure 1.** Twenty cortical brain areas with the highest degree of anatomical connectivity with the hippocampus. (**A**) Histogram plotting the mean structural connectivity (n=10; given by the sum of SIFT2-weighted values) associated with the 20 cortical areas most strongly connected with the whole hippocampus (excluding medial temporal lobe [MTL] areas; see *Figure 1—figure supplement 1* for MTL values). Error bars represent the standard error of the mean. (**B**) Histogram plotting the corresponding mean SIFT2-weighted values associated with anterior (yellow), body (red), and tail (blue) portions of the hippocampus for the 20 most strongly connected cortical areas presented in (**A**). Errors bars represent the standard error of the mean.

The online version of this article includes the following figure supplement(s) for figure 1:

**Figure supplement 1.** Medial temporal lobe (MTL) cortices anatomical connectivity with the hippocampus.

each cortical area and their associated strength of connectivity with the whole hippocampus. A full list of all cortical areas of the HCPMMP and their associated strengths of connectivity is provided in *Supplementary file 1* .

## Do cortical areas display preferential connectivity along the anterior–posterior axis of the hippocampus?

Next, we conducted a more detailed characterisation of SC between each cortical area and the head, body, and tail portions of the hippocampus. For brevity, we present results relating to the 20 most highly connected cortical areas described above. Results are summarised in *Figure 1B* and *Supplementary file 4*, which lists each cortical area, their associated strength of connectivity with the head, body, and tail portions of the hippocampus, and the results of statistical analyses (Bonferroni-corrected paired-samples *t*-tests; see 'Materials and methods'). A full list of all cortical areas and their associated strengths of connectivity with the head, body, and tail portions of the hippocampus are provided in *Supplementary file 2*.

Each of the 20 most highly connected cortical areas displayed preferential connectivity with specific regions along the anterior–posterior axis of the hippocampus. These can be categorised into four distinct patterns: (i) an anterior-to-posterior gradient of increasing connectivity; (ii) a posterior connectivity bias; (iii) an anterior connectivity bias; and (iv) a body connectivity bias.

In total, 8 of the 20 most highly connected cortical areas displayed a gradient of increasing connectivity from the head to tail of the hippocampus. These were areas ProS, V1, V2, V3, DVT, V4, V6,

**Table 1.** Twenty cortical brain areas (excluding medial temporal lobe) with the highest degree of anatomical connectivity with the whole hippocampus.

| Cortical area | Location of area | Whole hippocampus | | |
|---|---|---|---|---|
| | | Mean SIFT2-weighted value (connectivity strength; n=10) | Standard error of mean | Percent of all cortical connections accounted for by area (percent of cortical connections excluding MTL areas) |
| TF | Lateral temporal cortex | 7673 | 886 | 5.10 (10.61) |
| ProS | Medial parietal cortex (including posterior cingulate) | 5483 | 784 | 3.64 (7.58) |
| V1 | Early visual cortex (occipital) | 5385 | 579 | 3.58 (7.45) |
| V2 | Early visual cortex (occipital) | 3840 | 462 | 2.55 (5.31) |
| POS1 | Medial parietal cortex (including posterior cingulate) | 3712 | 424 | 2.47 (5.13) |
| TGd | Temporal pole | 3465 | 288 | 2.30 (4.79) |
| TGv | Temporal pole | 3337 | 313 | 2.22 (4.61) |
| V3 | Early visual cortex (occipital) | 3079 | 450 | 2.05 (4.26) |
| TE2a | Lateral temporal cortex | 2214 | 288 | 1.47 (3.06) |
| VMV2 | Ventral stream visual cortex | 2105 | 247 | 1.40 (2.91) |
| RSC | Medial parietal cortex (including posterior cingulate) | 2063 | 121 | 1.37 (2.85) |
| VVC | Ventral stream visual cortex | 1956 | 220 | 1.30 (2.71) |
| DVT | Medial parietal cortex (including posterior cingulate) | 1939 | 335 | 1.29 (2.68) |
| POS2 | Medial parietal cortex (including posterior cingulate) | 1802 | 248 | 1.20 (2.49) |
| VMV1 | Ventral stream visual cortex | 1788 | 248 | 1.19 (2.47) |
| FFC | Ventral stream visual cortex | 1670 | 172 | 1.11 (2.31) |
| V4 | Early visual cortex (occipital) | 1601 | 127 | 1.06 (2.21) |
| TE2p | Lateral temporal cortex | 1288 | 198 | 0.86 (1.78) |
| V6 | Dorsal stream visual cortex | 1050 | 195 | 0.70 (1.45) |
| V3A | Dorsal stream visual cortex | 1029 | 197 | 0.68 (1.42) |

Column 1 displays cortical areas as defined by the Human Connectome Project Multi-Modal Parcellation (HCPMMP) scheme and ordered by strength of connectivity with the whole hippocampus (abbreviations for all cortical areas are defined in **Supplementary file 3**). Column 2 indicates the broader brain region within which each cortical area is located. Column 3 displays the mean SIFT2-weighted value (connectivity strength) associated with each brain area. Column 4 displays the standard error of the mean. Column 5 displays the percent of all cortical connections accounted for by each area. Values in brackets indicate the percent of cortical connections accounted for by each area when excluding medial temporal lobe (MTL) areas.

and V3A (abbreviations are defined in **Supplementary file 3**). The results of Bonferroni-corrected paired-samples *t*-tests revealed that these regions each showed a statistically significant difference in connectivity strength between the head and body, the body and tail, and the head and tail portions of the hippocampus (see **Figure 1B** and **Supplementary file 4**). That is, each of these cortical areas displayed the lowest connectivity with the hippocampal head, significantly increased connectivity with the body and the strongest connectivity with the tail.

Five areas had significantly stronger connectivity with the posterior 2/3 of the hippocampus. These were areas POS1, VMV1, VMV2, RSC, and POS2. The results of Bonferroni-corrected paired-samples *t*-tests revealed that these cortical areas displayed a high degree of connectivity with the body and tail of the hippocampus (no statistically significant difference in connectivity) and significantly lower connectivity with the hippocampal head (see **Figure 1B** and **Supplementary file 4**).

Also, 3 of the 20 most highly connected cortical areas had significantly greater connectivity with the anterior 2/3 of the hippocampus These were areas TF, TGd, and TGv. The results of Bonferroni-corrected paired-samples *t*-tests revealed that these cortical areas displayed a high degree of connectivity with the head and body of the hippocampus (no statistically significant difference in connectivity) and significantly lower connectivity with the tail of the hippocampus (see *Figure 1B* and *Supplementary file 4*).

Four areas had greater connectivity with the body of the hippocampus. These areas were TE2a, VVC, FFC, and TE2p. The results of Bonferroni-corrected paired-samples *t*-tests revealed that these areas showed statistically significant differences in connectivity strength between the head and body of the hippocampus. TE2a, VVC, and TE2p showed statistically significant differences in connectivity strength between the body and tail portions but FFC did not (following Bonferroni correction). TE2a also showed a statistically significant difference between the head and tail portions of the hippocampus but VVC, FFC, and TE2p did not (see *Figure 1B* and *Supplementary file 4*). That is, each of these cortical areas displayed the highest degree of connectivity with the body of the hippocampus.

To summarise, our results detail the degree to which specific cortical areas preferentially connect along the anterior–posterior axis of the hippocampus. While some cortical areas displayed gradients of connectivity strength along the anterior–posterior axis of the hippocampus, others displayed preferential connectivity with specific portions of the hippocampus.

## Do cortical areas display unique distributions of endpoint density within the hippocampus?

Our tractography pipeline was specifically tailored to allow streamlines to enter/leave the hippocampus and quantitatively measure the location and density of streamline endpoints within the hippocampus using TDI. We created endpoint density maps (EDMs) that allowed us to visualise the spatial distribution of hippocampal endpoint density associated with each cortical area (described in 'Materials and methods'). While it is not feasible to present the results for all cortical areas of the HCPMMP, we describe results for the 20 most highly connected cortical areas described above. In relation to nomenclature, our use of the term 'medial' hippocampus refers to inferior portions of the hippocampus aligning with the distal subiculum, presubiculum, and parasubiculum. Our use of the term 'lateral' hippocampus refers to inferior portions of the hippocampus aligning with the proximal subiculum and CA1. In instances that we refer to portions of the hippocampus that align with the DG/CA4 or CA3/2, we state these regions explicitly by name.

The results of group-level analyses confirmed that specific cortical areas preferentially connect with different regions within the human hippocampus. For example, areas in the medial parietal cortex (ProS, POS1, RSC, DVT, POS2) displayed high endpoint density primarily in medial portions of the posterior hippocampus (see yellow arrows in *Figure 2A* for a representative example of endpoint densities associated with RSC; see *Figure 2—figure supplement 2* for other areas). In contrast, areas in temporopolar and inferolateral temporal cortex (TF, TGd, TGv, TE2a, TE2p) displayed high endpoint density primarily along the lateral aspect of the anterior 2/3 of the hippocampus and in a circumscribed region of the anterior medial hippocampus (see blue and white arrows, respectively, in *Figure 2B* for a representative example of endpoint densities associated with TGv; see *Figure 2—figure supplement 3* for other areas). Similar to areas in the medial parietal cortex, areas in the occipital cortices (V1–4, V6, V3a) displayed high endpoint density primarily in the posterior medial hippocampus and, to a lesser degree, in a circumscribed region of the anterior medial hippocampus (see yellow and white arrows, respectively, in *Figure 2C* for endpoint densities associated with V1; see *Figure 2—figure supplement 4* for other areas).

In parallel with these differences, specific regions within the hippocampus displayed high endpoint density for multiple cortical areas. For example, several medial parietal and occipital cortical areas displayed high endpoint density in the posterior medial hippocampus (yellow arrows in *Figure 2—figure supplements 2 and 4*). In contrast, several cortical areas in the temporal pole and inferolateral temporal lobe displayed high endpoint density in the anterior lateral hippocampus (blue arrows in *Figure 2—figure supplement 3*). Another cluster of high endpoint density in the anterior medial hippocampus (at the level of the uncal apex) was more broadly associated with specific areas in temporal, medial parietal, and occipital cortices (white arrows in *Figure 2—figure supplements 2–4*).

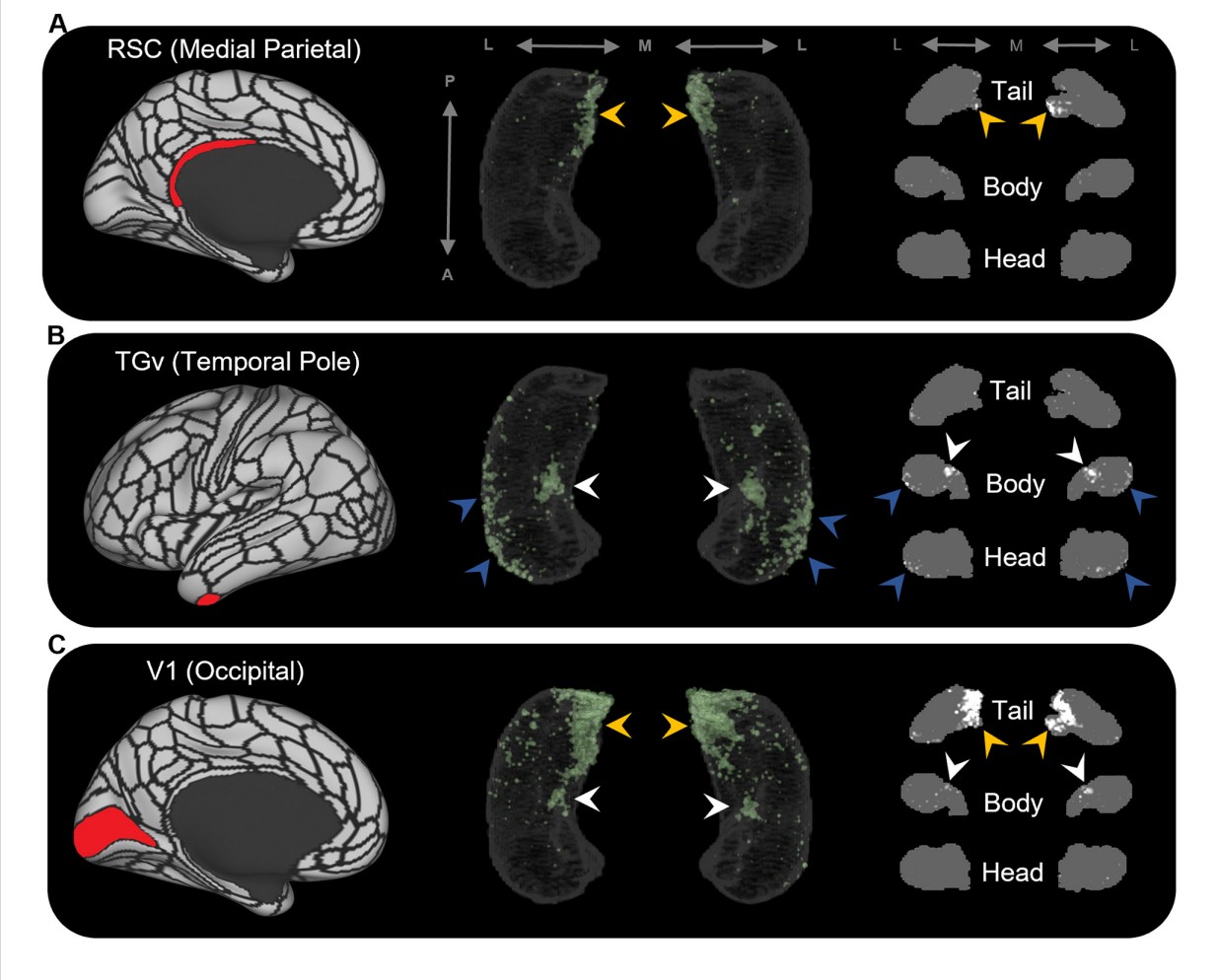

**Figure 2.** Representative examples of the spatial distribution of endpoint density within the hippocampus for different cortical brain areas. Representative examples of the location of endpoint densities associated with RSC in the medial parietal lobe (**A**), TGv in the temporal pole (**B**), and V1 in the occipital lobe (**C**). In each panel, the location of the relevant brain area is indicated in red on the brain map (left); a 3D-rendered representation of the bilateral group-level hippocampus mask is presented (middle; transparent grey) overlaid with the endpoint density map associated with each brain area (green); representative slices of the head, body, and tail of the hippocampus are displayed in the coronal plane (right; grey) and overlaid with endpoint density maps (white). Note that the spatial distribution of endpoint density within the hippocampus associated with each brain area differs along both the anterior–posterior and medial–lateral axes of the hippocampus. RSC and V1 displayed greatest endpoint density in the posterior medial hippocampus (yellow arrows in **A, C**). In contrast, TGv displayed greatest endpoint density in the anterior lateral hippocampus and in a circumscribed region in the anterior medial hippocampus (blue and white arrows, respectively, in **B**). Area V1 also expressed endpoint density in a circumscribed region in the anterior medial hippocampus (white arrows in **C**). A, anterior; P, posterior; M, medial; L, lateral.

The online version of this article includes the following figure supplement(s) for figure 2:

**Figure supplement 1.** Representative examples of the spatial distribution of endpoint density within the hippocampus for medial temporal lobe (MTL) brain areas.

**Figure supplement 2.** Representative examples of the spatial distribution of endpoint density within the hippocampus associated with the five most highly connected medial parietal brain areas.

**Figure supplement 3.** Representative examples of the spatial distribution of endpoint density within the hippocampus associated with the five most highly connected non-medial temporal lobe (non-MTL) temporal brain areas.

**Figure supplement 4.** Representative examples of the spatial distribution of endpoint density within the hippocampus associated with the five most highly connected occipital brain areas.

To further probe hippocampal endpoint density common to these cortical regions, we averaged the EDMs for anatomically related cortical areas. For example, we averaged the EDMs of the five most highly connected cortical areas in the temporal lobe (TF, TGd, TGv, TE2a, TE2p) and observed high endpoint density common to these areas was more clearly localised along the anterior lateral hippocampus and in a circumscribed region of the anterior medial hippocampus (blue and white arrows, respectively, in *Figure 3A*). Likewise, when we averaged the most highly connected cortical areas in the medial parietal and occipital cortices, respectively, high endpoint density common to each of these areas was localised in the posterior medial hippocampus and, to a lesser degree, in a circumscribed region of the anterior medial hippocampus (yellow and white arrows, respectively, in *Figure 3B and C*). When we averaged the EDMs across all of these areas, high endpoint density common to this broader collection was localised to separate circumscribed clusters in the posterior and anterior medial hippocampus (yellow and white arrows, respectively, in *Figure 3D*) and in punctate clusters along the anterior–posterior extent of the lateral hippocampus (blue arrows in *Figure 3D*). Our results suggest that these specific regions within the hippocampus are highly connected with multiple cortical areas in medial parietal, temporal, and occipital lobes.

In addition, our novel method allowed us to isolate and visualise streamlines between specific cortical areas and the hippocampus and map the spatial distribution of hippocampal endpoint density at the single-participant level. Representative examples are presented in *Figure 4*. *Figure 4B* displays isolated streamlines associated with areas TF and V1 in a single participant. *Figure 4D* displays the hippocampal EDMs associated with each of these cortical areas in the same participant. When overlaid on the T1-weighted image, EDMs resemble histological staining of postmortem tissue, albeit *in vivo* and at a coarser level of detail (see bottom panels of *Figure 4D*). For example, for area TF, in a coronal slice taken at the uncal apex (red line) endpoint density was primarily localised to specific areas in the medial hippocampus (red panel). For area V1, in a coronal slice taken at the hippocampal tail (turquoise line) endpoint density was also localised to the medial hippocampus (turquoise panel). When compared with equivalent sections of histologically stained tissue, the location of these clusters of endpoint density roughly aligns with the location of the distal subiculum/proximal presubiculum for both TF and V1 (indicated by black arrows; see also Figure 5A). The location of endpoint density associated with each cortical area was broadly consistent across participants (evidenced by the results of our group-level analyses).

Finally, while we observed clear overlaps in the group-averaged EDMs associated with specific cortical areas, a closer inspection of individual endpoints at the single-participant level revealed that endpoints associated with different cortical areas displayed both overlapping and spatially unique characteristics within these areas of overlap. For example, at the group level, areas V1 and V2 showed preferential connectivity with overlapping regions of the posterior medial hippocampus (see *Figure 2—figure supplement 4*) while, at the single-participant level, individual endpoints associated with each of these areas display both overlapping and spatially unique patterns (see *Figure 4— figure supplement 1*). This suggests that, while specific cortical areas display overlapping patterns of connectivity with specific regions of the hippocampus, subtle differences in how these cortical regions connect within these areas of overlap likely exist. A detailed examination of individual variability in these patterns, however, was beyond the scope of the current investigation and will be addressed in future studies.

## Discussion

This study represents a comprehensive *in vivo* characterisation of SC between cortical brain areas and the human hippocampus. We identified cortical areas with the highest degree of SC with the whole hippocampus, measured the degree to which these cortical areas preferentially connect along the anterior–posterior axis of the hippocampus, and deployed a tailored method to characterise where, within the hippocampus, each cortical area preferentially connects. Our results reveal how specific cortical areas preferentially connect with circumscribed regions along the anterior–posterior and medial–lateral axes of the hippocampus. Our results broadly reflect observations from the non-human primate literature (discussed below) and contribute new neuroanatomical insights to inform debates on human hippocampal function as it relates to its anterior–posterior axis. This work represents an important advance in our understanding of the neural architecture that underpins hippocampal-dependent memory systems in the human brain. In addition, our method represents a novel approach

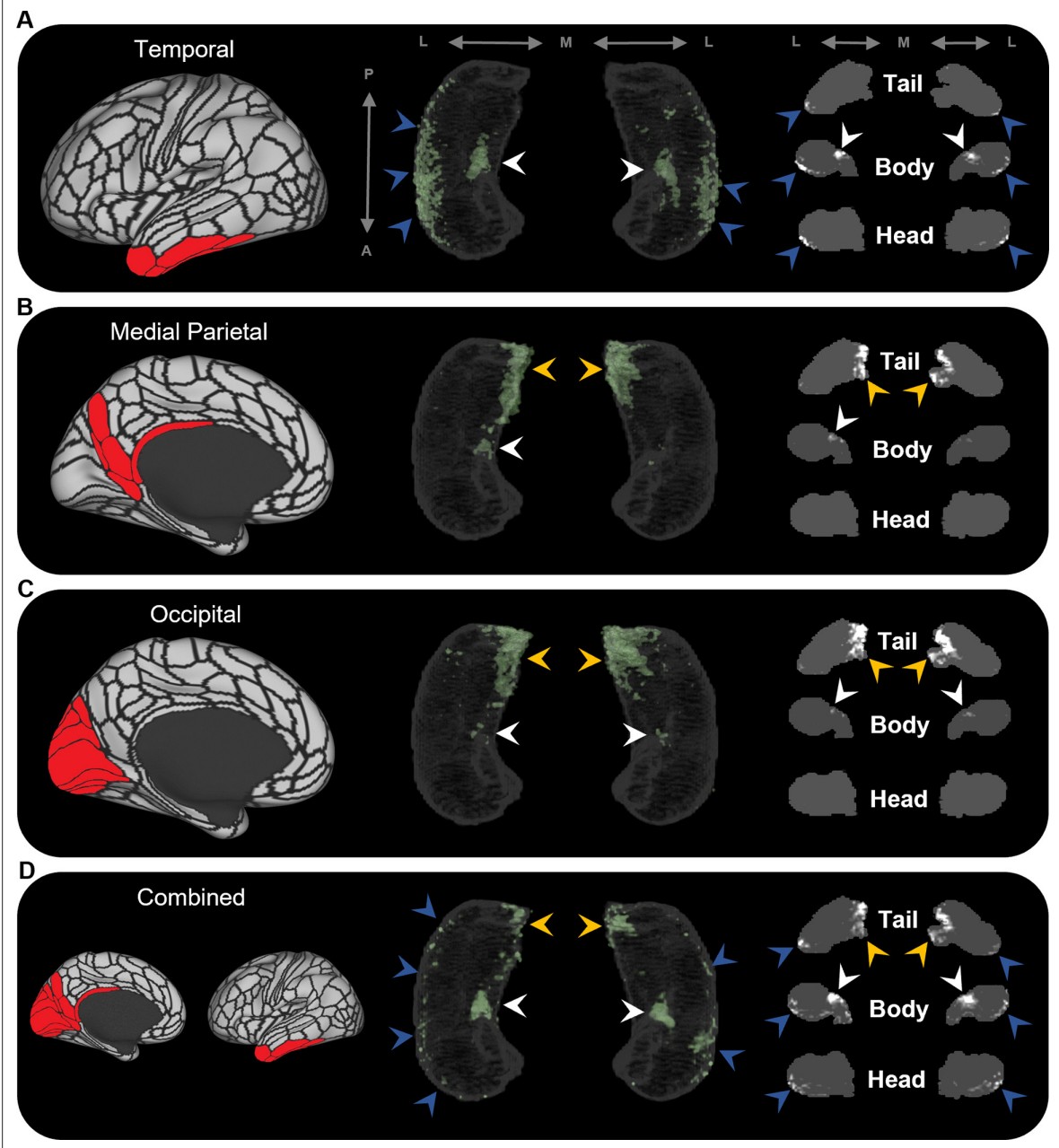

**Figure 3.** Averaged endpoint density maps for anatomically related brain areas. We averaged the endpoint density maps for the mostly highly connected brain areas in temporal (**A**; TF, TGd, TGv, TE2a, TE2p), medial parietal (**B**; ProS, POS1, RSC, DVT, POS2), and occipital (**C**; V1–4, V6, V3a) cortices and each of these regions combined (**D**). In each panel, the location of the relevant brain areas are indicated in red on the brain map (left); a 3D-rendered representation of the bilateral group-level hippocampus mask is presented (middle; transparent grey) overlaid with the endpoint density map associated with each collection of brain areas; representative slices of the head, body, and tail of the hippocampus are displayed in the coronal plane (right; grey) and overlaid with endpoint density maps (white). Average endpoint density associated with temporal areas (**A**) was primarily localised along the anterior lateral hippocampus and a circumscribed region in the anterior medial hippocampus (blue and white arrows, respectively). Average endpoint density associated with medial parietal (**B**) and occipital (**C**) areas was primarily localised to the posterior medial hippocampus (yellow arrows) and, to a lesser degree, circumscribed regions in the anterior medial hippocampus (white arrows). Average endpoint density associated with these temporal, medial parietal, and occipital brain areas combined (**D**) was localised to circumscribed regions in the posterior and anterior medial hippocampus (yellow and white arrows, respectively) and in punctate clusters along the anterior–posterior extent of the lateral hippocampus (blue arrows), suggesting that these specific regions within the hippocampus are highly connected with multiple cortical areas. A, anterior; P, posterior; M, medial; L, lateral.

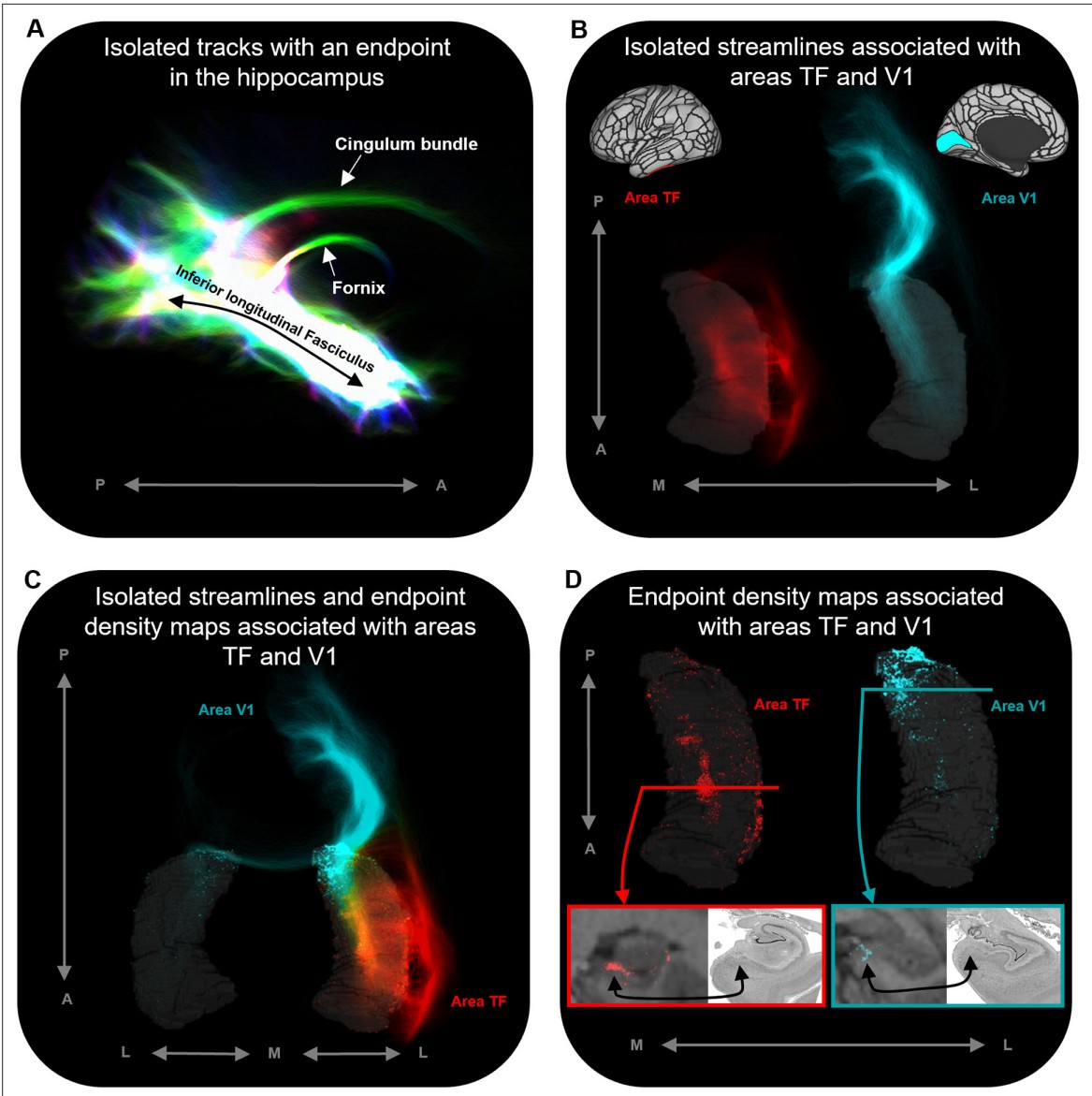

**Figure 4.** Representative examples of single-subject analysis. (**A**) 3D rendering of the hippocampus tractogram for a single participant showing isolated tracks with an endpoint in the hippocampus viewed in the sagittal plane (displayed with transparency; high intensity represents high density of tracks). (**B**) 3D-rendered left hippocampus masks (transparent grey) for the same participant overlaid with isolated streamlines associated with the left hemisphere areas TF (red) and V1 (turquoise). The location of areas TF and V1 is indicated on the brain maps (top). (**C**) 3D-rendered bilateral hippocampus mask for the same participant (transparent grey) overlaid with isolated streamlines and endpoint density maps associated with the left hemisphere areas TF (red) and V1 (turquoise). Note that, while streamlines associated with areas TF and V1 are primarily ipsilateral in nature, streamlines associated with V1 also project to the contralateral hippocampus. (**D**) 3D-rendered left hippocampus masks (transparent grey) for the same participant overlaid with endpoint density maps associated with areas TF (red) and V1 (turquoise). For TF and V1, we present a coronal section of the T1-weighted structural image overlaid with the endpoint density maps and a corresponding slice of postmortem hippocampal tissue (from a different subject) for anatomical comparison (bottom). For both TF (red border; level of the uncal apex) and V1 (turquoise border; level of the hippocampal tail), endpoint density is primarily localised to a circumscribed region in the medial hippocampus aligning with the location of the distal subiculum/proximal presubiculum (black arrows; also see *Figure 5A*). A, anterior; P, posterior; M, medial; L, lateral.

The online version of this article includes the following figure supplement(s) for figure 4:

**Figure supplement 1.** Cortical areas display both overlapping and spatially unique endpoints within specific regions of the hippocampus.

to conduct detailed *in vivo* investigations of the anatomical connectivity of the human hippocampus with implications for basic and clinical neuroscience.

## Preliminary whole-hippocampus and anterior–posterior axis analyses

First, we consider the preliminary whole-hippocampus analysis. Aligning with our predictions, the hippocampus was most strongly connected with the EC and highly connected with surrounding MTL structures (see 'Supplementary materials' for further information relating to MTL structures). Beyond the MTL, specific cortical areas in temporopolar, inferotemporal, medial parietal, and occipital cortices displayed the highest degree of SC with the hippocampus. These results broadly align with recent DWI investigations that reported SC between the whole hippocampus and these cortical regions in the human brain (*Maller et al., 2019*). Our anterior–posterior axis analyses provided a more detailed quantitative characterisation of cortico-hippocampal SC by measuring the degree to which specific cortical areas preferentially connect along the anterior–posterior axis of the hippocampus. In brief, specific areas within temporopolar and inferolateral temporal cortices displayed preferential SC with the head and/or body of the hippocampus. In contrast, medial parietal and occipital cortical areas displayed a posterior hippocampal connectivity bias most strongly with the hippocampal tail. These patterns of SC mirror commonly observed functional links between the anterior hippocampus and temporal regions and between the posterior hippocampus and parietal/occipital regions (*Dalton et al., 2019b*; *Tang et al., 2020*; *Poppenk and Moscovitch, 2011*; *Adnan et al., 2016*; *Barnett et al., 2021*). Our results provide new insights into the neuroanatomical architecture of these functional associations in the human brain. Further interpretation of these observations are facilitated by the results of our more detailed endpoint density analyses and are discussed below.

While many of our observed anatomical connections dovetail nicely with known functional associations, patterns of anatomical connectivity strength did not always mirror well-characterised functional associations between the hippocampus and cortical areas. For example, a surprising observation from our study was that only weak patterns of anatomical connectivity were observed between the hippocampus and the ventromedial prefrontal cortex (vmPFC) and other frontal cortical areas. This lies in contrast to well-documented functional associations between these regions (*Adnan et al., 2016*; *Barnett et al., 2021*; *Monk et al., 2021*). Our observation, however, supports a growing body of evidence that direct anatomical connectivity between the hippocampus and areas of the PFC may be surprisingly sparse in the human brain. For example, *Rosen and Halgren, 2022* recently reported that long-range connections between the hippocampus and functionally related frontal cortical areas may constitute fewer than 10 axons/mm$^2$ and more broadly observed that axon density between spatially distant but functionally associated brain areas may be much lower than previously thought. Our observation of sparse anatomical connectivity between the hippocampus and PFC mirrors this recent work and suggests a potential differentiation between structural and functional networks as they relate to the hippocampus. It remains possible, however, that methodological factors may contribute to these differences. We return to this point later in the discussion. A future dedicated study aimed at assessing whether the well-characterised functional associations between the hippocampus and vmPFC are driven by sparse direct connections or primarily by intermediary structures is necessary to address this issue in an appropriate level of detail.

## Endpoint density mapping of human cortico-hippocampal connectivity

To date, our knowledge of human cortico-hippocampal anatomical connectivity is largely inferred from the results of tract-tracing studies conducted in non-human primates and rodents. We know much less about these patterns in the human brain. To address this gap, we adapted a tractography pipeline to track streamlines entering/leaving the hippocampus, identify the location of their 'endpoints', and create spatial distribution maps of endpoint density within the hippocampus associated with each cortical area. The resulting EDMs allowed us to quantitatively assess and visualise where, within the hippocampus, different cortical areas preferentially 'connect'. To our knowledge, this is the first time such a specific approach has been used to map anatomical connectivity of the human hippocampus *in vivo*.

We observed striking differences in the location of endpoint density along both the anterior–posterior and medial–lateral axes of the hippocampus. For example, temporopolar and inferolateral temporal cortical areas displayed the greatest endpoint density along the lateral aspect of the head

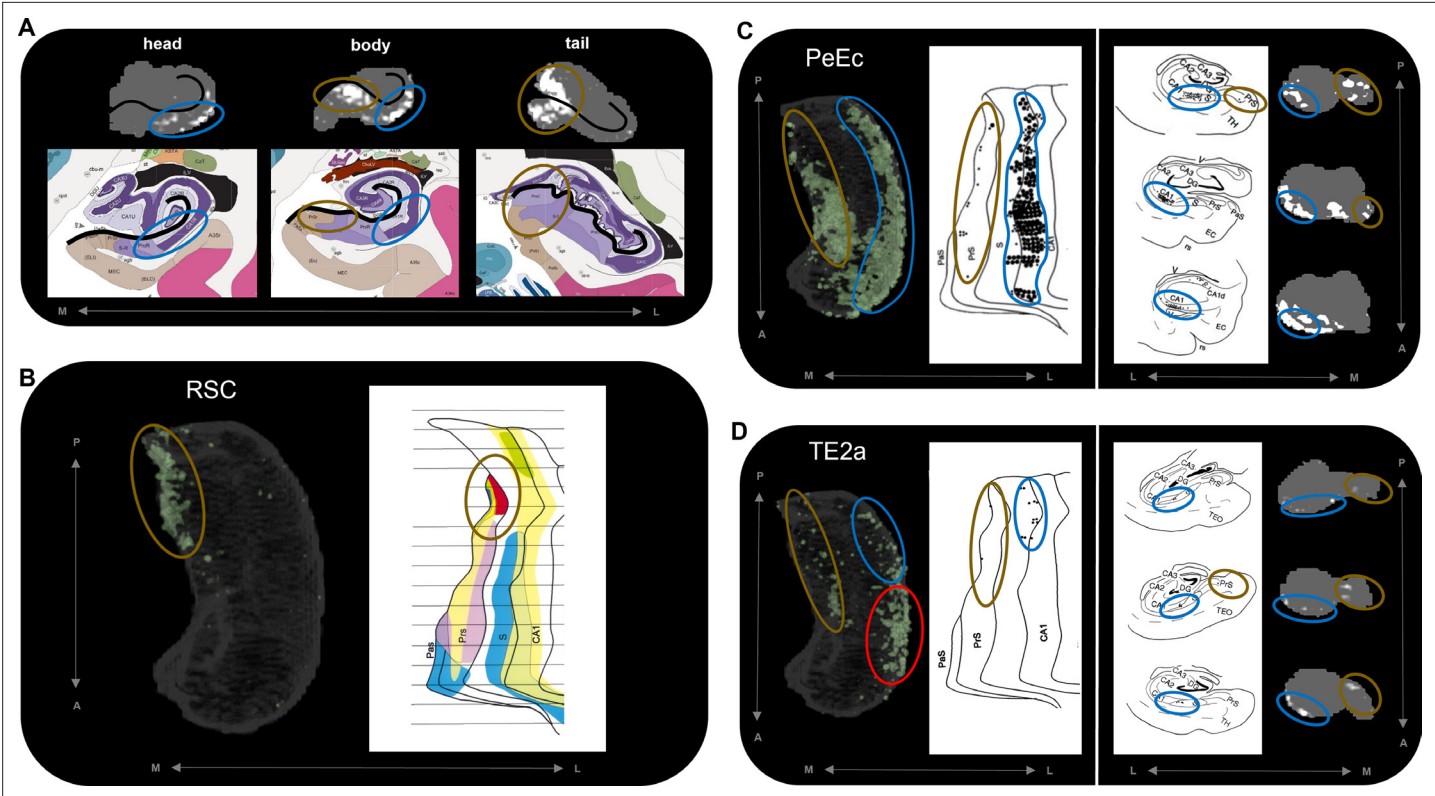

**Figure 5.** Anatomical location of endpoint densities within the hippocampus and comparison with results of non-human primate studies. (**A**) Representative slices of the head (left), body (middle), and tail (right) of the hippocampus displayed in the coronal plane (grey) and overlaid with group-level endpoint density maps associated with areas TF (head and body; white) and V1 (tail; white). Schematic representations of roughly equivalent slices of the hippocampus showing hippocampal subfields are displayed below each slice. Schematic representations were taken from the Allen Adult Human Brain Atlas website (https://atlas.brain-map.org/; *Ding et al., 2016*; *Allen Institute for Brain Science, 2004*). The vestigial hippocampal sulcus (black line) is overlaid on the hippocampus masks and schematic diagrams to aid comparison. Note that the endpoint density in the lateral hippocampus (blue ellipsoids) aligns with the location of the distal CA1/proximal subiculum. Endpoint density in the medial hippocampus (brown ellipsoids) aligns with the location of distal subiculum/proximal presubiculum. (**B–D**) 3D-rendered representations of the group-level hippocampus mask (left; transparent grey) are overlaid with endpoint density maps (green) associated with RSC (**B**), PeEc (**C**), and TE2a (**D**). Schematic representations of the macaque hippocampus (right; images reproduced with permission from *Insausti and Muñoz, 2001*) show the location of labelled cells following retrograde tracer injection into the RSC (**B**; red), PeEc (**C**; black points), and TE2a (**D**; black points) (*Insausti and Muñoz, 2001*). The right panel of (**C**) and (**D**) displays slices of the macaque hippocampus in the coronal plane displaying the location of labelled cells (black points) and roughly equivalent slices of human hippocampus in the coronal plane (grey) overlaid with endpoint density maps (white). Note that labelled cells and endpoint density in the macaque and human respectively are localised to similar regions along the anterior–posterior and medial (brown ellipsoids) – lateral (blue ellipsoids) axes of the hippocampus. However, areas of difference also exist (**D**; red ellipsoid). M, medial; L, lateral; A, anterior; P, posterior.

and body of the hippocampus (*Figure 2—figure supplement 3*; aligning with the location of distal CA1/proximal subiculum; see *Figure 5A*) and in a circumscribed cluster in the anterior medial hippocampus at the level of the uncal apex (*Figure 2—figure supplement 3*; aligning with the location of distal subiculum/proximal presubiculum; see *Figure 5A*). In contrast, areas in the medial parietal and occipital cortices displayed the greatest endpoint density along the medial aspect of the tail and body of the hippocampus (*Figure 2—figure supplement 2A–E* [medial parietal] and *Figure 2—figure supplement 4A–E* [occipital]; aligning with the location of the distal subiculum/proximal presubiculum; see *Figure 5A*). We discuss these observations in relation to the non-human primate literature below.

## Comparison to tract-tracing data in non-human primates

The results of our endpoint density analyses broadly overlap with observations from the non-human primate literature. For simplicity, we demonstrate this by comparing the results for three cortical areas with equivalent observations from the series of tract-tracing investigations of the macaque hippocampus described by *Insausti and Muñoz, 2001*. In the macaque, retrograde tracer injection into the RSC results in localised labelling of cells in the posterior presubiculum. Our results revealed high endpoint densities associated with the RSC were primarily localised to a homologous region in the posterior medial hippocampus (distal subiculum/proximal presubiculum; brown ellipsoids in *Figure 5B*). In the macaque, retrograde tracer injection into the PeEc results in dense labelling of cells along the anterior–posterior axis of the hippocampus primarily localised to the CA1/subiculum transition area and the presubiculum. Our results mirrored this pattern. We observed areas of high endpoint density along the lateral hippocampus (blue ellipsoids in *Figure 5C*; distal CA1/proximal subiculum) and along the medial hippocampus (brown ellipsoids in *Figure 5C*; distal subiculum/proximal presubiculum). In the macaque, labelling associated with area TE is scarce compared to that of the PeEc with labelled cells localised to the posterior CA1/subiculum transition area and the posterior presubiculum. Again, our results broadly aligned with this pattern. Compared with PeEc, we observed less endpoint density associated with TE2a (roughly corresponding to the injection site described by Insausti and Muñoz), which was primarily localised to the lateral (blue ellipsoids in *Figure 5D*; distal CA1/proximal subiculum) and medial (brown ellipsoids in *Figure 5D*; distal subiculum/proximal presubiculum) hippocampus. In contrast to the macaque, however, endpoint density was most strongly expressed in the lateral aspect of the anterior hippocampus (red ellipsoid in *Figure 5D*). We discuss potential interpretations for this and other differences later in the discussion. Overall, our results broadly aligned with patterns observed in the non-human primate brain and provide new and detailed insights regarding where specific cortical areas preferentially connect within the human hippocampus.

## New evidence for hubs of anatomical connectivity in the human hippocampus?

High endpoint density within the hippocampus was restricted to areas that aligned with the location of CA1, subiculum and the pre- and parasubiculum and was notably absent from areas aligning with DG/CA4 (hilus), CA3, and CA2. These observations mirror reports in the rodent and non-human primate literature where non-EC cortical areas predominantly connect with subicular cortices and the CA1/subiculum transition area (*Aggleton and Christiansen, 2015*; *Insausti and Muñoz, 2001*). Indeed, in non-human primates, the CA1/subiculum transition area and the presubiculum appear to be 'hotspots' of anatomical connectivity for multiple cortical areas (*Insausti and Muñoz, 2001*; *Kravitz et al., 2011*). In accordance with this, we observed that these specific regions within the human hippocampus displayed high endpoint density for multiple cortical areas. For example, the anterior lateral hippocampus (aligning with the distal CA1/proximal subiculum) displayed high endpoint density for multiple temporal cortical areas (*Figure 3A*) and the posterior medial hippocampus (aligning with the distal subiculum/proximal presubiculum) displayed high endpoint density for multiple areas of the medial parietal and occipital cortices (*Figure 3B and C*, respectively). Another cluster of high endpoint density in the anterior medial hippocampus (aligning with the distal subiculum/proximal presubiculum) was more broadly associated with temporal, medial parietal, and occipital areas (*Figure 3A–D*). Taken together, these observations provide evidence that discrete hubs of dense anatomical connectivity may exist along the anterior–posterior axis of the human hippocampus and that these hubs align with the location of the CA1/subiculum transition area and the distal subiculum/proximal presubiculum.

To the best of our knowledge, this is the first quantitative report of these patterns in the human brain and we highlight the intriguing possibility that two medial hippocampal hubs of high anatomical connectivity may exist; a posterior medial hub preferentially linked with visuospatial processing areas in medial parietal and occipital cortices and an anterior medial hub more broadly linked with temporopolar, inferotemporal, medial parietal, and occipital areas. We tentatively speculate that these circumscribed areas of the medial hippocampus could represent highly connected hubs of information flow between the hippocampus and distributed cortical networks. Indeed, the cortical areas identified in this study may represent key areas for direct cortico-hippocampal interactions in support of episodic/semantic memory processing and consolidation (*Nadel et al., 2000*).

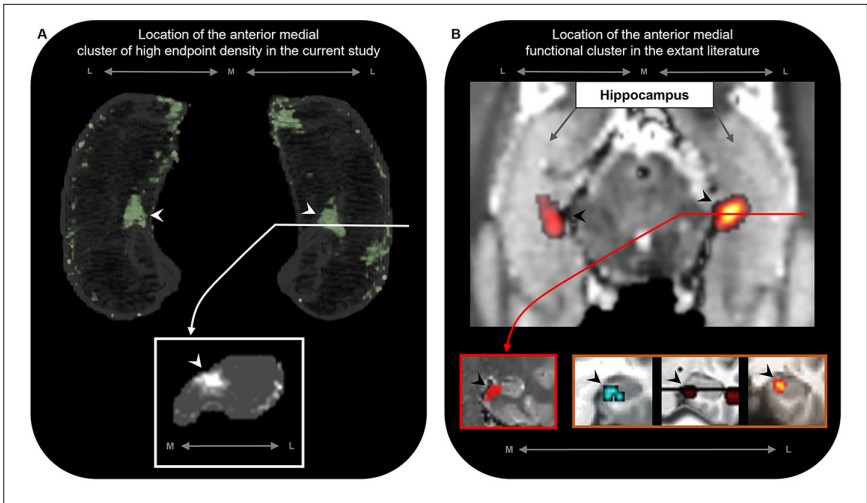

**Figure 6.** The location of the anterior medial anatomical cluster aligns with the location of a commonly observed anterior medial functional cluster. (**A**) A 3D-rendered representation of the bilateral group-level hippocampus mask (top; transparent grey) is presented overlaid with the endpoint density map averaged across the most highly connected brain areas in temporal, medial parietal, and occipital cortices (green; see *Figure 3D* for details); a representative slice of hippocampus in the coronal plane (bottom panel; grey) at the level of the uncal apex (indicated by white line) is presented overlaid with the endpoint density map (white). (**B**) An axial section of a T2-weighted image (top; from a separate study) showing the bilateral hippocampus overlaid with the location of a circumscribed functional cluster observed in the anterior medial hippocampus during a functional MRI investigation of visuospatial mental imagery, reproduced from Figure 2b from *Dalton et al., 2018*. A representative slice of the hippocampus (bottom-left panel; red border) at the level of the uncal apex (indicated by red line) is presented to show the location of this anterior medial functional cluster in the coronal plane, reproduced from Figure 3a from *Dalton et al., 2018*. Circumscribed functional clusters in the anterior medial hippocampus are commonly observed in studies of 'scene-based visuospatial cognition' such as episodic memory, prospection and scene perception (bottom-right panel; orange border). Left image was reproduced from Figure 3 from *Zeidman et al., 2015b*. Middle image was reproduced from Figure 3 from *Addis et al., 2012*. Right image was reproduced from Figure 4a from *Lee et al., 2013*. Note that the location of these commonly observed functional clusters in the anterior medial hippocampus (black arrows in panel **B**) aligns with the location of the anatomical cluster in the anterior medial hippocampus observed in this study (white arrows in panel **A**). M, medial; L, lateral.

Accumulating evidence from neuroimaging and clinical studies suggests that the medial hippocampus plays an important role in visuospatial cognition. The anterior medial hippocampus is consistently engaged during cognitive tasks that require processing of naturalistic scene stimuli in aid of episodic memory (*Addis et al., 2012*), visuospatial mental imagery (*Dalton et al., 2018*), and perception and mental construction (e.g. imagination) of scenes (*Zeidman et al., 2015b*; *Lee et al., 2013*). Strikingly, the location of the anterior medial cluster of anatomical connectivity observed in this study aligns with the location of functional clusters commonly observed in functional MRI investigations of these cognitive processes (*Dalton et al., 2018*; *Addis et al., 2012*; *Zeidman et al., 2015b*; *Lee et al., 2013*; *Zeidman and Maguire, 2016*; see *Figure 6* for comparison). Furthermore, recent evidence from the clinical domain suggests that the posterior medial hippocampus may be a critical hub in a broader memory circuit (*Ferguson et al., 2019*). The medial hippocampus has more broadly been proposed as a putative hippocampal hub for visuospatial cognition (*Dalton and Maguire, 2017*). Our results lend further support to these proposals by showing that specific regions of both the anterior

and posterior medial hippocampus display dense anatomical connectivity with multiple cortical areas in the human brain.

In addition, our results provide new anatomical insights to inform current debates on functional differentiation along the anterior–posterior axis of the human hippocampus. Our anterior–posterior axis analyses revealed that specific cortical areas displayed a gradient style increase in connectivity strength along the anterior–posterior axis of the hippocampus while others displayed non-linear patterns of connectivity. In parallel, the results of our endpoint density analyses suggest that discrete clusters of dense connectivity may also exist within the hippocampus. Together, these observations provide new evidence to support recent proposals that both gradients and circumscribed parcels of extrinsic connectivity may exist along the anterior–posterior axis of the human hippocampus (*Plachti et al., 2019*; *Brunec et al., 2018*; *Przeździk et al., 2019*; *Poppenk, 2020*; *Strange et al., 2014*) and that different circuits may exist within the hippocampus, each associated with different cortical inputs that underpin specific cognitive functions (*Dalton et al., 2018*). How the complex patterns of anatomical connectivity observed in this study relate to functional differentiation along the long axis of the hippocampus (*Plachti et al., 2019*; *Brunec et al., 2018*; *Przeździk et al., 2019*; *Poppenk, 2020*; *Strange et al., 2014*) and its subfields (*Dalton et al., 2019b*; *Dalton et al., 2019a*) will be a fruitful area of research in coming years.

Our results also have implications for the use of existing hippocampal subfield segmentation protocols as they relate to the anatomical connectivity of the human hippocampus. The patterns of anatomical connectivity we observed in this study may not map well to classical definitions of subfield boundaries currently used in MRI investigations of human hippocampal subfields (*Dalton et al., 2017*; *Berron et al., 2017*; *Iglesias et al., 2015*). For example, the anterior medial clusters of high endpoint density observed in this study appear to extend across the distal subiculum and proximal presubiculum and lateral clusters appear to extend across the proximal subiculum and distal CA1. This mirrors recent observations in the functional literature that suggest functional clusters also extend across classically defined subfield boundaries (*Dalton et al., 2019a*; *Grande et al., 2022*). Indeed, more fine-grained segmentation can reveal that results, initially attributed to a specific subfield, may actually be driven by a specific subportion within that subfield (*Dalton et al., 2019a*). Results of this study provide a neuroanatomical rationale for these observations and further support the notion that, in some contexts, it may be advantageous to eschew classical concepts of hippocampal subfields. Future studies will aim to assess how patterns of SC relate to patterns of functional connectivity within the hippocampus and, subsequently, inform decisions of how we structurally and functionally define hippocampal subfields in human MRI studies.

## Are there human-specific patterns of cortico-hippocampal connectivity?

Despite areas of concordance with the non-human primate literature noted above, we also observed important differences. As noted earlier, we observed broader patterns of endpoint density for area TE2a than we expected based on the non-human primate literature. Also, non-human primate studies have found direct and substantial connectivity between the hippocampus and orbitofrontal and superior temporal cortices (*Insausti and Muñoz, 2001*). We found only weak patterns of connectivity between the hippocampus and these regions. Also of note, we found denser patterns of anatomical connectivity between the posterior medial hippocampus and early visual processing areas in the occipital lobe than would be expected based on observations from the non-human primate literature. However, this observation supports recent reports of similar patterns of anatomical connectivity as measured by DWI in the human brain (*Maller et al., 2019*) and functional associations between these areas (*Tang et al., 2020*; *Dugré et al., 2021*). Collectively, these findings are potentially of great conceptual importance for how we think about the hippocampus and its connectivity with early sensory cortices in the human brain and open new avenues to probe the degree to which these regions may interact to support visuospatial cognitive functions such as episodic memory, mental imagery, and imagination and perhaps even more abstract cognitive functions relating to creativity. It is important to note that our methods differ significantly from the carefully controlled injection of tracer into circumscribed portions of the brain. MRI investigations of SC are inherently less precise, and methodological limitations likely explain some of our observed differences. It is equally important to note, however, that while we expect to see evolutionarily conserved overlaps in SC between macaque and human brains, we should not expect exactly the same patterns of connectivity. Since

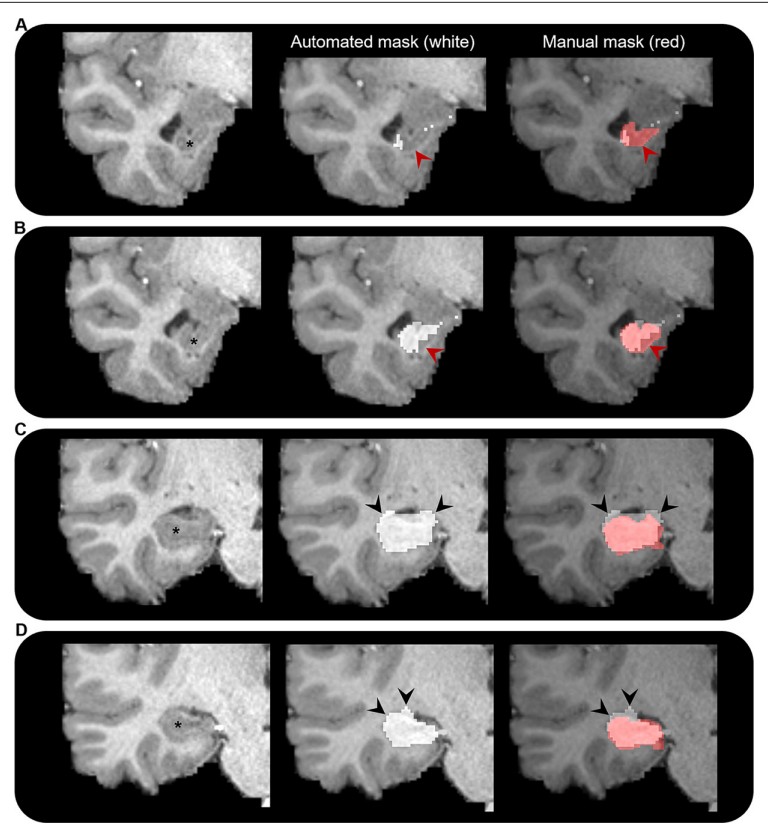

**Figure 7.** Comparison of automated and manual hippocampus segmentations. Representative examples of the automated hippocampus mask derived from the Human Connectome Project Multi-Modal Parcellation (HCPMMP) and the manually segmented hippocampus mask. We display examples from anterior (**A**) to posterior (**D**) portions of the hippocampal head. In each panel, we present a coronal slice of the T1-weighted image focused on the right temporal lobe for a single participant (left; hippocampus indicated by *), the same image overlaid with the automated hippocampus mask derived from the HCPMMP (middle; white) and the same image overlaid with both the automated HCPMMP hippocampus mask (right; white) and the manually segmented hippocampus mask (transparent red). Note that in the anterior-most slices (**A, B**) the automated mask does not cover the entire extent of the hippocampus (indicated by red arrows) and in more posterior slices (**C, D**) the automated mask often overextends across the lateral ventricle superior to the hippocampus and into the adjacent white matter (indicated by black arrows). Streamlines making contact with these erroneous portions of the automated hippocampus mask may lead to results that are biologically implausible.

The online version of this article includes the following figure supplement(s) for figure 7:

**Figure supplement 1.** Adjustment of grey matter–white matter interface (gmwmi) underlying the hippocampus.

**Figure supplement 2.** Analysis pipeline.

splitting from a common ancestor, macaques and humans have likely evolved species-specific patterns of cortico-hippocampal connectivity to support species-specific cognitive functions. Whether differences observed in this study reflect methodological limitations, species differences or perhaps most likely, a mix of both, require further investigation.

To our knowledge, only one prior study has attempted to characterise the broader hippocampal 'connectome' in the healthy human brain (*Maller et al., 2019*). Our study differs from this recent report in several important technical aspects. We analysed connectivity profiles of the head, body, and tail portions of the hippocampus separately in addition to the whole hippocampus. We manually delineated the hippocampus for each participant to ensure full coverage of the hippocampus and minimise the hippocampus mask 'spilling' into adjacent white matter (a common occurrence with automated segmentation methods; see *Figure 7*). We used the HCPMMP scheme, which provides more detailed subdivisions of cortical grey matter (*Glasser et al., 2016*). We also included SIFT2 (*Smith et al., 2015a*) in our analysis pipeline to increase biological accuracy of quantitative connectivity estimates.

Finally, we used a carefully adapted tractography method that incorporates anatomical constraints and allowed streamlines to enter/leave the hippocampus, which provided us with a means to determine the location and topography of streamline 'endpoints', and therefore their distribution within the hippocampus (see 'Materials and methods' for detail relating to each of these points).

In conclusion, this study represents a first attempt to apply this method and has some limitations. Specifically, our method relies on several manual steps to delineate the hippocampus and amend the gmwmi that abuts the inferior portion of the hippocampus (described in 'Materials and methods'). This can be time consuming and requires expertise to accurately identify the hippocampus along its entire anterior–posterior extent on structural MRI scans. However, considering automated methods of hippocampal segmentation are sometimes not sufficiently accurate, particularly in the anterior and posterior most extents of the hippocampus (see *Figure 7* for representative examples), manual delineation is the gold standard and ensures the best results. We restricted our analysis to a limited number of subjects under the age of 35 and selected participants whose hippocampus was clearly visible along its entire anterior–posterior axis on T1-weighted structural scans. While this ensured we used the best data quality available, further work should explore how these results may differ in the context of healthy ageing and in diseases that affect the hippocampus such as Alzheimer's disease, epilepsy, and schizophrenia. How reliable our pipeline is in data acquired with more traditional clinical protocols remains to be explored and was beyond the scope of this study. Fibre-tracking results are known to be greatly influenced by the particular algorithm implementation. To ensure the robustness of our results, we used state-of-the-art methods, which included a diffusion model that is robust to crossing fibres and presence of partial volume effect (*Jeurissen et al., 2014*), an advanced probabilistic tractography algorithm that incorporates anatomical priors (*Smith et al., 2012*), and a streamline to fibre density matching to improve its quantitative properties (*Smith et al., 2015a*). The interested reader is referred to a recent review article where these and other key issues of ensuring the reliability of fibre tracking are discussed (*Calamante, 2019*). Despite these advances and the high-quality HCP data used in this study, limitations in spatial resolution likely restrict our ability to track particularly convoluted white matter pathways within the hippocampus and our results should be interpreted with this in mind. Indeed, this may explain the surprising lack of endpoint density observed in the DG/CA4-CA3 regions of the hippocampus where we would expect to see high endpoint density associated with, for example, the EC, which is known to project to these regions. Future dedicated studies using higher resolution data are needed to assess these pathways in greater detail. Also, we cannot rule out that some connections observed in this study may result from limitations inherent to current probabilistic fibre-tracking methods whereby tracks can mistakenly 'jump' between fibre bundles (e.g. for connections between the posterior medial hippocampus and area V1 due to the proximity to the optic radiation), especially in 'bottleneck' areas. Again, future work using higher resolution data may allow more targeted investigations necessary to confirm or refute the patterns we observed here. These limitations notwithstanding, our results provide new detailed insights into anatomical connectivity of the human hippocampus, can inform theoretical models of human hippocampal function as they relate to the long axis of the hippocampus, and can help fine-tune network connectivity models.

It should also be noted that we did not attempt to map the connections outlined in this study to canonical white matter tracts. The reliable segmentation of white matter fibre bundles is currently an area of contention in the DWI community. This pervasive and problematic issue was highlighted in a recently published large multisite study that revealed a high degree of variability in how white matter bundles are defined, even from the same set of whole-brain streamlines (*Schilling et al., 2021*). This significantly limits meaningful comparison and/or interpretation. Indeed, such an approach may paradoxically take away from the detailed characterisations we have achieved in this study. As highlighted by *Schilling et al., 2021*, it is now paramount that consensus is reached to define criteria to reliably and reproducibly define white matter fibre bundles.

From a clinical perspective, the hippocampus is central to several neurodegenerative and neuropsychiatric disorders. Considering healthy memory function is dependent upon the integrity of white matter fibres that connect the hippocampus with the rest of the brain, developing a more detailed understanding of the anatomical connectivity of the anterior–posterior axis of the hippocampus and its subfields has downstream potential to help us better understand hippocampal-dependent memory decline in ageing and clinical populations. Our novel method can potentially be harnessed to measure changes in anatomical connectivity between the hippocampus and cortical areas known to be affected

in neurodegenerative diseases, to assist with monitoring of disease progression and/or as a diagnostic tool. In addition, our method could be deployed to visualise patient-specific patterns of hippocampal connectivity to support surgical planning in patients who require MTL resection for intractable MTL epilepsy.

# Materials and methods

## Participant details

Ten subjects (seven females) were selected from the minimally processed HCP 100 unrelated subject database (<35 years old). Subjects were selected based on the scan quality and visibility of the outer boundaries of the hippocampus on each participant's T1-weighted structural MRI scan. This was done in order to increase the anatomical accuracy of our hippocampal segmentations (described below).

## Image acquisition

The HCP diffusion protocol consisted of three diffusion-weighted shells (b-values: 1000, 2000, and 3000 s/mm$^2$, with 90 diffusion weighting directions in each shell) plus 18 reference volumes (b = 0 s/mm$^2$). Each diffusion-weighted image was acquired twice, with opposite phase-encoded direction to correct for image distortion (*Andersson et al., 2003*). The diffusion image matrix was 145 × 145 with 174 slices and an isotropic voxel size of 1.25 mm. The TR and TE were 5520 and 89.5 ms, respectively. Each subject also included a high-resolution T1-weighted dataset, which was acquired with an isotropic voxel size of 0.7 mm, TR/TE = 2400/2.14 ms, and flip angle = 8°.

## Manual segmentation of the hippocampus

The whole hippocampus was manually segmented for each participant on coronal slices of the T1-weighted image using ITK-SNAP (*Yushkevich et al., 2006*). Although automated methods of hippocampal segmentation are available, they are sometimes not sufficiently accurate particularly in the anterior and posterior-most extents of the hippocampus (see *Figure 7* for representative examples). Although labour intensive, manual segmentation by an expert in hippocampal anatomy remains the gold standard for detailed and accurate investigation of the human hippocampus. We adapted the manual segmentation protocol outlined by *Dalton et al., 2017*. While this protocol details a method for segmenting hippocampal subfields, we followed guidelines as they relate to the outer boundaries of the hippocampus. This ensured that the whole-hippocampus mask for each participant contained all hippocampal subfields (DG, CA4-1, subiculum, presubiculum, and parasubiculum) and encompassed the entire anterior–posterior extent of the hippocampus (see *Dalton et al., 2017* for details). Representative examples of the hippocampus mask are presented in *Figure 7* and *Figure 7—figure supplement 1*. Manual segmentations were conducted by an expert in human hippocampal anatomy and MRI investigation of the human hippocampus (MAD) with 14 years experience including histological (*Valenzuela et al., 2012*) and MRI investigations (*Dalton et al., 2018*; *Dalton et al., 2019b*; *Dalton et al., 2019a*; *Dalton et al., 2017*). We took particular care to ensure that all boundaries of the hippocampus mask (including inferior, superior, medial, and lateral aspects) did not encroach into adjacent white or grey matter structures (e.g. amygdala, thalamic nuclei). This minimised the potential fusion of white matter tracts associated with other areas with our hippocampus mask. For the anterior–posterior axis analysis, we split each participant's whole-hippocampus mask into thirds corresponding with the head, body, and tail of the hippocampus. This was done in accordance with commonly used anatomical landmark-based methods. In brief, the demarcation point between the head and body of the hippocampus was the uncal apex (*Poppenk et al., 2013*; *Zeidman et al., 2015a*), and the demarcation point between the body and tail of the hippocampus was the anterior-most slice in which the crus of the fornix was fully visible (*Kulaga-Yoskovitz et al., 2015*; *Bernasconi et al., 2003*). Although these landmarks are commonly used to divide the long axis of the hippocampus, it should be noted that these are somewhat arbitrary divisions based on gross anatomical landmarks and do not reflect specific anatomical, functional, or genetic boundaries within the hippocampus (*Strange et al., 2014*).

## Image preprocessing and whole-brain tractography

Besides the steps carried out by the HCP team as part of the minimally processed datasets, the additional image processing pipeline included in our analysis is summarised in *Figure 7—figure*

*supplement 2A*. Processing was performed using the MRtrix software package (http://www.mrtrix.org) (*Tournier et al., 2019*; *Tournier et al., 2012*). Additional processing steps were implemented in accordance with previous work (*Civier et al., 2019*) and included bias-field correction (*Tustison et al., 2010*) as well as multi-shell multi-tissue constrained spherical deconvolution to generate a fibre orientation distribution (FOD) image (*Jeurissen et al., 2014*; *Tournier et al., 2004*; *Tournier et al., 2007*). The T1 image was used to generate a 'five-tissue-type' (5TT) image using FSL (*Smith et al., 2012*; *Smith, 2002*; *Zhang et al., 2001*; *Patenaude et al., 2011*; *Smith et al., 2004*); tissue 1 = cortical grey matter, tissue 2 = sub-cortical grey matter, tissue 3 = white matter, tissue 4 = CSF, and tissue 5 = pathological tissue. The FOD image and the 5TT image were used to generate 70 million anatomically constrained tracks (*Smith et al., 2012*) using dynamic seeding (*Smith et al., 2015a*) and the second-order Integration over Fibre Orientation Distributions (iFOD2; *Tournier et al., 2010*) probabilistic fibre-tracking algorithm. The relevant parameters included 70 million tracks, dynamic seeding, backtracking option specified, FOD cutoff 0.06, minimum track length 5 mm, maximum track length 300 mm, and maximum of 1000 attempts per seed.

## Hippocampus tractography

We developed a tailored pipeline to track streamlines into the hippocampus. To do this, we first amended the gmwmi immediately inferior to the hippocampus. This was necessary because the manually segmented hippocampus mask lay slightly superior to the automatically generated gmwmi (see *Figure 7—figure supplement 1B*; middle image). Pilot testing showed that streamlines terminated when reaching this portion of the gmwmi, thereby impeding streamlines from traversing the inferior border of the hippocampus. This was a problem because white matter fibres innervate the hippocampus primarily through this region (and also via the fimbria/fornix; *Duvernoy, 2005*). It was, therefore, important to ensure that streamlines could cross the inferior border of the hippocampus mask in a biologically plausible manner. To facilitate this, we created an additional hippocampus mask for each participant that extended inferiorly to encompass portions of the gmwmi that lay immediately inferior to the hippocampus (see *Figure 7—figure supplement 1*; right image). This amended hippocampus mask was labelled as white matter in the modified 5TT image (referred to as m5TT). This served to remove the portion of the gmwmi immediately inferior to the hippocampus and ensured that streamlines could enter/leave the hippocampus in a biologically plausible manner. Additionally, the original whole-hippocampus segmentation was assigned as fifth tissue type in the m5TT image (i.e. where no anatomical priors are applied within the ACT framework in MRtrix; *Figure 7—figure supplement 2B*). This allowed streamlines to move within the hippocampus.

In summary, amending the erroneous gmwmi allowed streamlines to traverse hippocampal boundaries in a biologically plausible manner and labelling the manually segmented hippocampus as a fifth tissue type permitted streamlines to move within the hippocampus. Together, this allowed us to follow the course of each streamline within the hippocampus and determine the location of each streamline 'endpoint' (described below).

Next, the FOD image was used with the m5TT image to generate an additional 10 million tracks. This set of anatomically constrained tracks (*Smith et al., 2012*) was seeded from the manually segmented hippocampus, and iFOD2 was used for fibre tracking (*Tournier et al., 2010*). The 70 million whole-brain tracks and the 10 million hippocampus tracks were combined, and spherical-deconvolution informed filtering of tractograms 2 (SIFT2; *Smith et al., 2015a*) was used on the combined 80 million track file, thereby assigning a weight to each track and providing biological credence to the connectivity measurements (*Smith et al., 2015b*). Within the SIFT2 framework, connectivity is then computed not by counting the number of tracks but by the sum of its SIFT2 weights. Tracks (and SIFT2 weights) that had an endpoint in the hippocampus were extracted (referred to here as the 'hippocampus tractogram') and used in both the whole-hippocampus and anterior–posterior axis analyses.

## Whole hippocampus connectivity

FreeSurfer (*Fischl, 2012*) was used to further process the T1-weighted image. The HCPMMP 1.0 (*Glasser et al., 2016*) was mapped to each subject in accordance with previous work (*Tahedl, 2020*). The parcellation divided the cerebral cortex into 360 parcels (180 per hemisphere). Importantly, we replaced the automated hippocampus and presubiculum parcels with the manually segmented hippocampus (which included the presubiculum) for greater anatomical accuracy (referred to as 'modified

HCPMMP'; *Figure 7—figure supplement 2C*). The SC of tracks between the hippocampus and the other parcels was obtained using tracks (and SIFT2 weights) from the hippocampus and the modified HCPMMP (containing the whole-hippocampus segmentation). The strength of connectivity between the hippocampus and every other parcel of the HCPMMP was measured by the sum of the SIFT2-weighted connectivity values (*Smith et al., 2015a*). For each parcel (cortical area), we combined left and right hemisphere values (i.e. left and right RSC) and report bilateral results.

### Anterior–posterior axis connectivity

As described in the manual segmentation section, the whole hippocampus was subsequently divided into thirds (head, body, and tail, as shown in *Figure 7—figure supplement 2D*). For the anterior–posterior axis analyses, each of these three regions were added to the HCPMMP as their own unique parcel. In a similar manner to the whole-hippocampus connectivity analysis, the strength of connectivity between each hippocampal region (head, body, and tail) and each parcel of the modified HCPMMP was measured by the sum of the SIFT2-weighted connectivity values. To assess whether connectivity values for each of the top 20 most highly connected cortical brain areas significantly differed between the head, body, and tail portions of the hippocampus, we conducted Bonferroni-corrected paired-samples $t$-tests for each area. We conducted three tests for each cortical brain area; head vs. body; body vs. tail; and head vs. tail. These are reported in the main text when significant at an adjusted p-value of <0.016.

### TDI mapping of tracks between the whole hippocampus and the other parcels and endpoint creation

The extracted hippocampus tractogram was used to isolate tracks (and weights) between the whole-hippocampus parcel and every other parcel in the modified HCPMMP file. Two different TDI maps (*Calamante et al., 2012*; *Calamante et al., 2010*) were computed for each parcel; a TDI of the hippocampus tractogram and a TDI map showing only the endpoints of this tractogram. Both TDI maps were constructed at 0.2 mm isotropic resolution. These TDI endpoint maps were used in the group-level analysis described below. Note that we refer to these TDI endpoint maps as 'endpoint density maps' (EDMs) in the main text.

### Group-level analysis - group-level hippocampus template and TDI endpoint map registration

We employed the symmetric group-wise normalization method (SyGN) (*Avants et al., 2010*) implemented in the ANTs toolbox (http://stnava.github.io/ANTs/; *Cook, 2022*) to build a population-specific hippocampus template (*Lv et al., 2022*). Specifically, the cross-correlation metric was used to optimise the boundary agreement among the hippocampi masks of each participant. Then, each individual hippocampus mask was registered to the generated population template with the combined linear and non-linear transformation. For each participant, the transformation parameters were recorded and applied to the TDI endpoint maps which were warped into the template space at a resolution of 0.7 mm isotropic. The group average was then calculated by providing a group-level distribution map of endpoint density within the hippocampus for each parcel of interest. EDMs were visualised in mrview (the MRtrix image viewer). Representative images displayed in our figures were visualised with the minimum and maximum intensity scale set at 0 and 0.05, respectively, and a minimum and maximum threshold set at 0.02 and 0.5, respectively.

## Acknowledgements

Data were provided by the Human Connectome Project, WU-Minn Consortium (principal investigators: David Van Essen and Kamil Ugurbil; 1U54MH091657) funded by the 16 NIH Institutes and Centers that support the NIH Blueprint for Neuroscience Research; and by the McDonnell Center for Systems Neuroscience at Washington University, St. Louis, MO.

We are grateful for the support of the National Health and Medical Research Council of Australia (grant numbers APP1091593 and APP1117724) and the Australian Research Council (grant number DP170101815). The authors acknowledge the technical assistance provided by the Sydney Informatics Hub and Sydney Imaging, two Core Research Facilities of the University of Sydney, Australia.

## Additional information

### Competing interests

Fernando Calamante: is listed as one of the inventors in a patent awarded for the track-density imaging method. The other authors declare that no competing interests exist.

### Funding

| Funder | Grant reference number | Author |
|---|---|---|
| National Health and Medical Research Council | APP1091593 | Fernando Calamante |
| National Health and Medical Research Council | APP1117724 | Fernando Calamante |
| Australian Research Council | DP170101815 | Fernando Calamante |

The funders had no role in study design, data collection and interpretation, or the decision to submit the work for publication.

### Author contributions

Marshall A Dalton, Conceptualization, Data curation, Formal analysis, Validation, Investigation, Visualization, Methodology, Writing – original draft, Project administration, Writing – review and editing; Arkiev D'Souza, Data curation, Formal analysis, Methodology, Writing – review and editing; Jinglei Lv, Formal analysis, Investigation, Methodology, Writing – review and editing; Fernando Calamante, Conceptualization, Resources, Supervision, Funding acquisition, Investigation, Methodology, Project administration, Writing – review and editing

### Author ORCIDs

Marshall A Dalton http://orcid.org/0000-0003-4089-6801
Arkiev D'Souza http://orcid.org/0000-0002-5932-2042
Jinglei Lv http://orcid.org/0000-0002-4906-2646
Fernando Calamante http://orcid.org/0000-0002-7550-3142

### Ethics

We used fully anonymised data from the Human Connectome Project (HCP) 100 unrelated subject database; https://www.humanconnectome.org/study/hcp-young-adult/data-releases.

### Decision letter and Author response

Decision letter https://doi.org/10.7554/eLife.76143.sa1
Author response https://doi.org/10.7554/eLife.76143.sa2

## Additional files

### Supplementary files

• Transparent reporting form

• Supplementary file 1. Connectivity values between the cortical mantle and the whole hippocampus. Connectivity values between the whole hippocampus and all cortical areas in the Human Connectome Project Multi-Modal Parcellation (HCPMMP) scheme are presented. Column 1 displays each broad brain region ordered by their strength of connectivity with the whole hippocampus. Column 2 displays the percent of all cortical connections accounted for by each region. Values in brackets indicate the percent of cortical connections accounted for by each region when excluding medial temporal lobe (MTL) areas. Column 3 presents the cortical areas located within each broad brain region ordered by their strength of connectivity with the whole hippocampus (abbreviations for all cortical areas are defined in *Supplementary file 3*). Column 4 displays the mean SIFT2 weighted value (connectivity strength) associated with each cortical area. Column 5 displays the standard error of the mean. Column 6 displays the percent of all cortical connections accounted for by each cortical area. Values in brackets indicate the percent of cortical connections accounted for by each cortical area when excluding MTL areas. Column 7 displays the

rank order for each cortical area by strength of connectivity. Values in brackets indicate the rank order when excluding MTL areas.

• Supplementary file 2. Connectivity between the cortical mantle and the head, body, and tail of the hippocampus. Connectivity values between the head, body, and tail of the hippocampus and all cortical brain areas included in the Human Connectome Project Multi-Modal Parcellation (HCPMMP) scheme are presented. Column 1 displays each cortical area ordered by strength of connectivity with the whole hippocampus (abbreviations for all cortical areas are defined in *Supplementary file 3*). Column 2 presents the broader brain region within which each cortical area is located. Column 3 displays the mean SIFT2-weighted value (connectivity strength) associated with each cortical area and the head of the hippocampus. Column 4 displays the associated standard error of the mean. Column 5 displays the mean SIFT2-weighted value (connectivity strength) associated with each cortical area and the body of the hippocampus. Column 6 displays the associated standard error of the mean. Column 7 displays the mean SIFT2-weighted value (connectivity strength) associated with each cortical area and the tail of the hippocampus. Column 8 displays the associated standard error of the mean.

• Supplementary file 3. List of abbreviations for all cortical brain areas in the Human Connectome Project Multi-Modal Parcellation (HCPMMP) scheme.

• Supplementary file 4. Connectivity between the 20 most highly connected cortical areas and the head, body, and tail of the hippocampus. Column 1 displays cortical areas as defined by the Human Connectome Project Multi-Modal Parcellation (HCPMMP) scheme and ordered by strength of connectivity with the whole hippocampus (abbreviations for all cortical areas are defined in *Supplementary file 3*). Column 2 designates the portion of hippocampus (head, body, tail). Column 3 displays the mean SIFT2-weighted value (connectivity strength) between each cortical area and the head, body, and tail of the hippocampus. Column 4 displays the standard error of the mean. Column 5 displays the contrast for each paired-samples *t*-test. Column 6 displays the *t*-statistic associated with each pair. Column 7 displays the p-value associated with each pair. Column 8 indicates the significance level associated with each pair following Bonferroni correction. ***<0.001, **<0.01, *<0.05, ns, not statistically significant.

## Data availability

We used data from the Human Connectome Project (HCP) 100 unrelated subject database; https://www.humanconnectome.org/study/hcp-young-adult/data-releases. All data needed to evaluate the conclusions in the paper are present in the paper and/or the Appendix. Code used for our analyses are available via github at the following address; https://github.com/Marshall-Dalton/Pipeline_for_anatomical_connectivity_along_the_anterior-posterior_axis_of_the_human_hippocampus (copy archived swh:1:rev:c341129cf85fb5345321b60c7ec76faa858698fb).

The following previously published dataset was used:

| Author(s) | Year | Dataset title | Dataset URL | Database and Identifier |
|---|---|---|---|---|
| Van Essen DC, Ugurbil K, Auerbach E, Barch D, Behrens TEJ, Bucholz R, Chang A, Chen L, Corbetta M, Curtiss SW, Della Penna S, Feinberg D, Glasser MF, Harel N, Heath AC, Larson-Prior L, Marcus D, Michalareas G, Moeller S, Oostenveld R, Petersen SE, Prior F, Schlaggar BL, Smith SM, Snyder AZ, Xu J, Yacoub E | 2017 | 1200 Subjects Data Release | https://www.humanconnectome.org/study/hcp-young-adult/document/1200-subjects-data-release | Human Connectome Project, 1200-subjects-data-release |

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

# Appendix 1

## Additional medial temporal lobe analyses

While the primary focus of our study was to characterise anatomical connectivity between the hippocampus and non-MTL cortical areas, we also characterised patterns of connectivity between the hippocampus and MTL cortical areas. Cortical areas within the MTL were highly connected with the hippocampus and cumulatively accounted for 52% of all cortical connections. The most highly connected area was the EC (24% of all cortical connections) followed by PeEc (14%), PHA2 (6%), PHA1 (5%), and PHA3 (3%) (see *Figure 1—figure supplement 1A* and *Supplementary file 1*). Results of the anterior–posterior axis analyses revealed that the EC and PeEc displayed a gradient-style anterior-to-posterior decrease in connectivity. In contrast, PHA1-3 each displayed the highest degree of connectivity with the body of the hippocampus and lower connectivity with both the head and tail of the hippocampus (*Figure 1—figure supplement 1B* and *Supplementary file 2*).

The results of the group-level endpoint analyses showed that MTL cortical areas had dense patterns of connectivity with the hippocampus. For example, the EC displayed high endpoint density along the entire anterior–posterior axis of the hippocampus. Although difficult to visualise due to the high density of endpoints, the results of our anterior–posterior analysis showed that endpoint density was greatest in the head and body of the hippocampus (see *Figure 1—figure supplement 1B* and *Supplementary file 2*). Endpoint density was primarily located in portions of the hippocampus aligning with the location of the CA1, subiculum, and pre- and parasubiculum (see *Figure 2—figure supplement 1A*).

Endpoint density expression associated with PeEc differed along the anterior–posterior axis of the hippocampus. In the hippocampal head, endpoint density was more pronounced in the lateral hippocampus primarily aligning with the location of the CA1 and subiculum (indicated by blue arrows in *Figure 2—figure supplement 1B*). Moving into the body, expression in the CA1 and subiculum was maintained (indicated by blue arrows in *Figure 2—figure supplement 1B*) and additional clusters of endpoint density were expressed in the medial hippocampus aligning with the location of the distal subiculum/proximal presubiculum (indicated by white arrows in *Figure 2—figure supplement 1B*). Moving into the hippocampal tail, clusters of endpoint density appeared to be more localised to a lateral region of the hippocampus aligning with the location of distal CA1/proximal subiculum (indicated by blue arrows in *Figure 2—figure supplement 1B*) with comparatively weaker expression in medial portions of the hippocampal tail.

Endpoint density associated with PHA1-3 (numbered from medial to lateral in the posterior parahippocampal cortex) was most pronounced in the body of the hippocampus and primarily localised to lateral areas aligning with the location of CA1 and subiculum (indicated by blue arrows in *Figure 2—figure supplement 1C–E*) and in the medial hippocampus aligning with the location of the distal subiculum/proximal presubiculum (indicated by white arrows in *Figure 2—figure supplement 1C–E*). PHA1-3 expressed less endpoint density in the hippocampal head, which was localised to the lateral hippocampus aligning with the location of CA1 and subiculum (indicated by blue arrows in *Figure 2—figure supplement 1C–E*). Moving into the hippocampal tail, clusters of endpoint density were more localised to a lateral region aligning with the location of distal CA1/proximal subiculum (indicated by blue arrows in *Figure 2—figure supplement 1C–E*). However, PHA1-3 showed different patterns of endpoint density in the medial aspect of the hippocampal tail. PHA1 and, to a lesser extent, PHA2 displayed endpoint density in the posterior medial hippocampus aligning with the location of the distal subiculum/proximal presubiculum (indicated by yellow arrows in *Figure 2—figure supplement 1C and D*). In contrast, PHA3 displayed modest endpoint density in the posterior medial hippocampus (*Figure 2—figure supplement 1E*).

