## [Editor Report]

This is an historical paper that is methodologically exceptional that offers new insights into the heterogeneity of human hippocampal anatomical pathways. Coming at a time when functional studies and theoretical papers are recognizing that this heterogeneity is of critical importance in furthering our understanding of hippocampal function, this paper will provide a nice guide for researchers in their ongoing hypothesis testing. Congratulations on an invaluable contribution!

---

## [Decision Letter]

**Decision letter after peer review:**

Thank you for submitting your article "Anatomical connectivity along the anterior-posterior axis of the human hippocampus – new insights using quantitative fibre-tracking" for consideration by *eLife*. Your article has been reviewed by 2 peer reviewers, and the evaluation has been overseen by a Reviewing Editor and Timothy Behrens as the Senior Editor. The following individual involved in the review of your submission has agreed to reveal their identity: Bryan Strange (Reviewer #2).

Essential revisions:

The reviewers and reviewing editor strongly agree that the results presented herein are novel and have great potential to influence ongoing and future work examining hippocampal-cortical networks. As you can see from the reviews, the paper was well-received but both reviewers noted some places where more information would be tremendously useful. Reviewer 2 points out the surprising V1 connectivity with posterior hippocampus. This has a lot of potential for impact but it would be important to rule out more convincingly that nearby visual thalamic regions, specifically the LGN, is not somehow contributing to this result. Likewise, the lack of a strong medial PFC cortical target is also very interesting. Could this be partly due to the fiber tracts being smaller (the uncinate fasciculus?) or is it truly multisynaptic? Please address these two points with further presentation of existing results or new data, if possible.

In addition, please address the major points, and wherever possible, the minor ones of these two stellar reviews as I believe they have done an outstanding job, as experts in the field, enumerating the kinds of questions our expert readers will also share.

*Reviewer #1 (Recommendations for the authors):*

I just have a few additional suggestions for clarifications and data presentation.

1. The theoretical motivation is clear and well set up. The paper would be strengthened by the inclusion of a few more methodological details in the Introduction to provide a high-level description of what the present method and approach accomplish over and above previous investigations. This would also make it easier to compare the present results with previous findings.

2. The figures are detailed and very clear and the results are easy to follow. I would potentially suggest grouping the bar plots in Figure 1B based on their connectivity profiles outlined on p. 8 (lines 172-175) to clearly highlight how the regions differ. But I leave it up to the authors to decide whether to implement this change, I appreciate that it could also make the figure more confusing.

3. For the purpose of clarity, it might be helpful to include a figure or panel presenting the results at the level of networks of regions – i.e., the left-most column in Table S1. This might provide an intuitive overview of the broad patterns, making it more accessible collapsing across the individual region labels.

*Reviewer #2 (Recommendations for the authors):*

1. In view of the potential conceptual importance of the V1 finding for how we think about the hippocampus as receiving stronger inputs from higher order association cortex over primary sensory cortex, I encourage the authors to try to exclude tracts that pass along the other margins of the hippocampus but are known to connect other structures.

2. One way to avoid the medial-lateral terminology confusion would have been to attempt subfield segmentation in these 10 cases.

---

## [Author Response]

Essential revisions:The reviewers and reviewing editor strongly agree that the results presented herein are novel and have great potential to influence ongoing and future work examining hippocampal-cortical networks. As you can see from the reviews, the paper was well-received but both reviewers noted some places where more information would be tremendously useful. Reviewer 2 points out the surprising V1 connectivity with posterior hippocampus. This has a lot of potential for impact but it would be important to rule out more convincingly that nearby visual thalamic regions, specifically the LGN, is not somehow contributing to this result. Likewise, the lack of a strong medial PFC cortical target is also very interesting. Could this be partly due to the fiber tracts being smaller (the uncinate fasciculus?) or is it truly multisynaptic? Please address these two points with further presentation of existing results or new data, if possible.

Many thanks. We are grateful for the Reviewers and Reviewing Editors positive comments. We address the points noted above directly in our response to Reviewers.

In addition, please address the major points, and wherever possible, the minor ones of these two stellar reviews as I believe they have done an outstanding job, as experts in the field, enumerating the kinds of questions our expert readers will also share.

We address each point below and greatly appreciate the time taken by the Reviewers to comment on our work. We believe their suggestions have greatly improved the quality of our manuscript.

Reviewer #1 (Recommendations for the authors):I just have a few additional suggestions for clarifications and data presentation.1. The theoretical motivation is clear and well set up. The paper would be strengthened by the inclusion of a few more methodological details in the Introduction to provide a high-level description of what the present method and approach accomplish over and above previous investigations. This would also make it easier to compare the present results with previous findings.

We have now included a brief description in the Introduction highlighting the key methodological advances used in the current study.

We have included the following text in the Introduction (Pages 4-5, lines 130-144);

“In typical fibre-tracking studies, we cannot reliably ascertain where streamlines would naturally terminate, as they have been found to also display unrealistic terminations, such as in the middle of white matter or in cerebrospinal fluid (39). While methods have been proposed to ensure more meaningful terminations (40), for example, with terminations forced at the grey matter-white matter interface (gmwmi), this approach is still not appropriate for characterising terminations within complex structures like the hippocampus. A key methodological advance of our approach was to remove portions of the gmwmi inferior to the hippocampus (where white matter fibres are known to enter/leave the hippocampus). This allowed streamlines to permeate the hippocampus in a biologically plausible manner. Importantly, we combined this with a tailored processing pipeline that allowed us to follow the course of streamlines within the hippocampus and identify their ‘natural’ termination points. These simple but effective methodological advances allowed us to map the spatial distribution of streamline ‘endpoints’ within the hippocampus. We further combined this approach with state-of-the-art tractography methods that incorporate anatomical information (40) and assign weights to each streamline (41) to achieve quantitative connectivity results that more faithfully reflect the biological accuracy of the connection’s strength (39).”

2. The figures are detailed and very clear and the results are easy to follow. I would potentially suggest grouping the bar plots in Figure 1B based on their connectivity profiles outlined on p. 8 (lines 172-175) to clearly highlight how the regions differ. But I leave it up to the authors to decide whether to implement this change, I appreciate that it could also make the figure more confusing.

We appreciate this suggestion from Reviewer 1. We did amend Figure 1B in alignment with this recommendation but the authors concluded that these changes made the figure more confusing (as the reviewer themselves noted it may). Our preference is, therefore, to keep figure 1B in its original format.

3. For the purpose of clarity, it might be helpful to include a figure or panel presenting the results at the level of networks of regions – i.e., the left-most column in Table S1. This might provide an intuitive overview of the broad patterns, making it more accessible collapsing across the individual region labels.

We see the potential utility of this suggestion from Reviewer 1 but, after contemplation, we have ultimately decided that we disagree with the suggestion for the following reason. We feel that presenting connectivity patterns collapsed across region labels will diminish an important aspect of our results; namely that the hippocampus seems to be highly connected with discrete cortical areas within these broad regions rather than uniformly connected with the region as a whole. For example, the medial parietal cortex was split into 14 cortical areas but just 5 of these accounted for 89% of all connections between the medial parietal cortex and hippocampus. We fear that including a figure that collapses results across any particular region (i.e., medial parietal cortex) will provide potentially misleading representations of the more discrete patterns of connectivity described in this manuscript. We believe, therefore, that the visual depiction of this more discrete network of highly connected cortical regions presented in Figure 3, is a more accurate and appropriate representation of our results.

Reviewer #2 (Recommendations for the authors):1. In view of the potential conceptual importance of the V1 finding for how we think about the hippocampus as receiving stronger inputs from higher order association cortex over primary sensory cortex, I encourage the authors to try to exclude tracts that pass along the other margins of the hippocampus but are known to connect other structures.

We agree with Reviewer 2 that our observations relating to area V1 could be the result of limitations inherent to current tracking methodology. Indeed, probabilistic tracking can result in tracks mistakenly ‘jumping’ between fibre bundles. Unfortunately, primarily due to limitations in image resolution, we do not believe that we can categorically rule this possibility out in the current dataset beyond the measures we have already taken in our analysis pipeline. We have now included additional text in the Discussion acknowledging and emphasising this possible limitation of our study.

We have included the following text in the Discussion section (Page 26; Lines 727-732);

“Also, we cannot rule out that some connections observed in the current study may result from limitations inherent to current probabilistic fibre-tracking methods whereby tracks can mistakenly ‘jump’ between fibre bundles (e.g. for connections between the posterior medial hippocampus and area V1 due to the proximity to the optic radiation), especially in “bottleneck” areas. Again, future work using higher resolution data may allow more targeted investigations necessary to confirm or refute the patterns we observed here.”

These points notwithstanding, our results support recently observed structural and functional associations between the posterior hippocampus and early visual processing areas. We agree that these findings are potentially of great conceptual importance for how we think about the hippocampus and its connectivity with primary sensory cortices in the human brain and we have now included a brief comment relating to this in the Discussion.

We have included the following text in the Discussion (Pages 24; Lines 656-662);

“However, this observation supports recent reports of similar patterns of anatomical connectivity as measured by DWI in the human brain (38) and functional associations between these areas (43, 60). Collectively, these findings are potentially of great conceptual importance for how we think about the hippocampus and its connectivity with early sensory cortices in the human brain and open new avenues to probe the degree to which these regions may interact to support visuospatial cognitive functions such as episodic memory, mental imagery and imagination.”

2. One way to avoid the medial-lateral terminology confusion would have been to attempt subfield segmentation in these 10 cases.

We agree that applying the terms ‘medial’ and ‘lateral’ to our three-dimensional representations can lead to some ambiguities and confusion. We have included a new description defining our use of these terms in the Results section.

We have included the following text in the Results section (Page 10; Lines 269-274).

“In relation to nomenclature, our use of the term ‘medial’ hippocampus refers to inferior portions of the hippocampus aligning with the distal subiculum, presubiculum and parasubiculum. Our use of the term ‘lateral’ hippocampus refers to inferior portions of the hippocampus aligning with the proximal subiculum and CA1. In instances that we refer to portions of the hippocampus that align with the DG or CA3/2 we state these regions explicitly by name”.

In relation to attempting subfield segmentation, our results suggest that patterns of endpoint density do not map to classically defined hippocampal subfield boundaries. For example, anterior medial clusters of high endpoint density appear to extend across the distal subiculum and proximal presubiculum and lateral clusters appear to extend across the proximal subiculum and distal CA1. Considering these clusters extend across hippocampal subfield boundaries, we suggest that it may be advantageous to eschew classically defined subfield boundaries in order to report patterns inherent to the data. In line with this, we believe a more restrictive formal subfield analysis may, again, paradoxically take away from the detailed characterisations we have achieved in the current study. A dedicated study is needed to characterise, in detail, how the patterns of endpoint density observed in the current study align with classically defined hippocampal subfield boundaries currently used in MRI investigations. Such an analysis was beyond the scope of the current study and will be addressed with future work.

On reflection, we believe this is a very important discussion point that should have been included in our original manuscript and we thank Reviewer 2 for focussing our minds in this direction. We have, therefore, included the following text in the Discussion section (Page 23; Lines 631-645) which we believe greatly improves our manuscript.

“Our results also have implications for the use of existing hippocampal subfield segmentation protocols as they relate to the anatomical connectivity of the human hippocampus. The patterns of anatomical connectivity we observed in the current study may not map well to classical definitions of subfield boundaries currently used in MRI investigations of human hippocampal subfields (59-61). For example, the anterior medial clusters of high endpoint density observed in the current study appear to extend across the distal subiculum and proximal presubiculum and lateral clusters appear to extend across the proximal subiculum and distal CA1. This mirrors recent observations in the functional literature that suggest functional clusters also extend across classically defined subfield boundaries (17, 62). Indeed, more fine-grained segmentation can reveal that results, initially attributed to a specific subfield, may actually be driven by a specific sub-portion within that subfield (17). Results of the current study provide a neuroanatomical rationale for these observations and further support the notion that, in some contexts, it may be advantageous to eschew classical concepts of hippocampal subfields. Future studies will aim to assess how patterns of structural connectivity relate to patterns of functional connectivity within the hippocampus and, subsequently, inform decisions of how we structurally and functionally define hippocampal subfields in human MRI studies.”